# Global-scale distribution of ozone in the remote troposphere from ATom and HIPPO airborne field missions.

Ilann Bourgeois[1,2], Jeff Peischl[1,2], Chelsea R. Thompson[1,2], Kenneth C. Aikin[1,2], Teresa Campos[3], Hannah Clark[4], Róisín Commane[5], Bruce Daube[6], Glenn W. Diskin[7], James W. Elkins[8], Ru-Shan Gao[2], Audrey Gaudel[1,2], Eric J. Hintsa[1,8], Bryan J. Johnson[8], Rigel Kivi[9], Kathryn McKain[1,8], Fred L. Moore[1,8], David D. Parrish[1,2], Richard Querel[10], Eric Ray[1,2], Ricardo Sánchez[11], Colm Sweeney[8], David W. Tarasick[12], Anne M. Thompson[13], Valérie Thouret[14], Jacquelyn C. Witte[3], Steve C. Wofsy[6], and Thomas B. Ryerson[2].

[1]Cooperative Institute for Research in Environmental Sciences, University of Colorado Boulder, Boulder, CO, USA

[2]NOAA CSL, Boulder, CO, USA

[3]National Center for Atmospheric Research, Boulder, CO, USA

[4]IAGOS-AISBL, Brussels, Belgium

[5]Department of Earth and Environmental Sciences, Lamont-Doherty Earth Observatory of Columbia University, New York, NY, USA

[6]School of Engineering and Applied Sciences, Harvard University, Cambridge, MA, USA

[7]NASA Langley Research Center, Hampton, VA, USA

[8]NOAA GML, Boulder, CO, USA

[9]Finnish Meteorological Institute, Space and Earth Observation Centre, Sodankylä, Finland

[10]National Institute of Water & Atmospheric Research (NIWA), Lauder, NZ

[11]Servicio Meteorológico Nacional, Buenos Aires, Argentina

[12]Experimental Studies Research Division, MSC/Environment and Climate Change Canada, Downsview, Ontario, CA

[13]Earth Sciences Division, NASA/Goddard Space Flight Center, Greenbelt, MD, USA

[14]Laboratoire d'Aérologie, CNRS and Université Paul Sabatier, Université de Toulouse, Toulouse, FR

**Abstract**

Ozone is a key constituent of the troposphere where it drives photochemical processes, impacts air quality, and acts as a climate forcer. Large-scale in situ observations of ozone commensurate with the grid resolution of current Earth system models are necessary to validate model outputs and satellite retrievals. In this paper, we examine measurements from the Atmospheric Tomography (ATom, 4 deployments in 2016–2018) and the HIAPER Pole-to-Pole Observations (HIPPO; 5 deployments in 2009–2011) experiments, two global-scale airborne campaigns covering the Pacific and Atlantic basins.

ATom and HIPPO represent the first global-scale, vertically resolved measurements of $O_3$ distributions throughout the troposphere, with HIPPO sampling the atmosphere over the Pacific and ATom sampling both the Pacific and Atlantic. Given the relatively limited temporal resolution of these two campaigns, we first compare ATom and HIPPO ozone data to longer-term observational records to establish the representativeness of our dataset. We show that these two airborne campaigns captured on average 53, 54, and 38 % of the ozone variability in the marine boundary layer, free troposphere, and upper troposphere/lower stratosphere (UTLS), respectively, at nine well-established ozonesonde sites. Additionally, ATom captured the most frequent ozone concentrations measured by regular commercial aircraft flights in the northern Atlantic UTLS. We then use the repeated vertical profiles from these two campaigns to confirm and extend the existing knowledge ~~global-scale picture~~ of tropospheric ozone spatial and vertical distributions throughout the remote troposphere. We highlight a clear hemispheric gradient, with greater ozone in the northern hemisphere, consistent with greater precursor emissions and consistent with previous modeling and satellite studies. We also show that the ozone distribution below 8 km was similar in the extra-tropics of the Atlantic and Pacific basins, likely due to zonal circulation patterns. However, twice as much ozone was found in the tropical Atlantic than in the tropical Pacific, due to well-documented dynamical patterns transporting continental air masses over the Atlantic. Finally, we show that the seasonal variability of tropospheric ozone over the Pacific and the Atlantic basins is driven year-round by transported continental plumes and photochemistry, and the vertical distribution is driven by photochemistry and mixing with stratospheric air. This new dataset provides additional constraints for global climate and chemistry models to improve our understanding of both ozone production and loss processes in remote regions, as well as the influence of anthropogenic emissions on baseline ozone.

## 1. Introduction

Tropospheric ozone ($O_3$) plays a major role in local, regional, and global air quality and significantly influences Earth's radiative budget (IPCC, 2013; Shindell et al., 2012). In addition, $O_3$ drives tropospheric photochemical processes by controlling hydroxyl radical (OH) abundance, which subsequently controls the lifetime of other pollutants including volatile organic compounds (VOCs), methane, and some stratospheric ozone-depleting substances (Crutzen, 1974; Levy, 1971). Sources of $O_3$ to the troposphere include downward transport from the stratosphere (Junge, 1962) and photochemical production from precursors such as carbon monoxide (CO), methane ($CH_4$), and VOCs in the presence of nitrogen oxides ($NO_x$) from natural or anthropogenic sources (Monks et al., 2009). Tropospheric $O_3$ sinks include photo-dissociation, chemical reactions, and dry deposition. Owing to its relatively long lifetime (~23 days in the troposphere; Young et al., 2013), $O_3$ can be transported across hemispheric scales. $O_3$ mixing ratios over a region thus depend not only on local and regional sources and sinks, but also on long-range transport. Further, the uneven density of $O_3$ monitoring locations around the globe leads to significant sampling gaps, especially near developing nations and away from land (Gaudel et al., 2018). The troposphere over the remote oceans is among the least-sampled regions, despite hosting 60–70 % of the global tropospheric $O_3$ burden (Holmes et al., 2013).

Since the early 1980's, several aircraft campaigns have addressed this paucity of remote observations, most notably under the umbrella of the Global Tropospheric Experiment (GTE), a major component of the National Aeronautics and Space Administration (NASA) Tropospheric Chemistry Program (https://eosweb.larc.nasa.gov/project/gte/gte_table). Airborne campaigns have targeted both the Pacific and Atlantic Oceans, providing novel characterization of $O_3$ sources, distribution, and photochemistry in the marine troposphere (Browell et al., 1996a; Davis et al., 1996; Jacob et al., 1996; Pan et al., 2015; Schultz et al., 1999; Singh et al., 1996c) and the low-$O_3$ tropical Pacific pool (Singh et al., 1996b), the pervasive role of continental outflow on $O_3$ production (Bey et al., 2001; Crawford et al., 1997; Heald et al., 2003; Kondo et al., 2004; Martin et al., 2002; Zhang et al., 2008), and the marked influence of African and South American biomass burning on $O_3$ production in the Southern Hemisphere (Browell et al., 1996b; Fenn et al., 1999; Mauzerall et al., 1998; Singh et al., 1996a; Thompson et al., 1996). Ozonesondes have been launched from remote sites for more than three decades in some places, and have provided additional constraints on the sources and photochemical balance of

tropospheric $O_3$, including a deep understanding of the vertically-resolved tropospheric $O_3$
climatology in select locations (Derwent et al., 2016; Diab et al., 2004; Jensen et al., 2012; Kley
et al., 1996; Liu et al., 2013; Logan, 1985; Logan and Kirchhoff, 1986; Newton et al., 2017;
Oltmans et al., 2001; Parrish et al., 2016; Sauvage et al., 2006; Thompson et al., 2012). Spatially-
resolved $O_3$ climatology has been provided by routine sampling by commercial aircraft, but has
mostly been limited to the upper troposphere or over continental regions (Clark et al., 2015;
Cohen et al., 2018; Logan et al., 2012; Petetin et al., 2016; Sauvage et al., 2006; Thouret et al.,
1998; Zbinden et al., 2013), and by satellite observations (Edwards et al., 2003; Fishman et al.,
1990, 1991; Hu et al., 2017; Thompson et al., 2017; Wespes et al., 2017; Ziemke et al., 2005,
2006, 2017), somewhat tempered by large uncertainties (Tarasick et al., 2019b). Recent
overview analyses depict the current understanding of global tropospheric $O_3$ sources,
distribution, and photochemical balance and underscore the insufficiency of observations in the
remote free troposphere (Cooper et al., 2014; Gaudel et al., 2018; Tarasick et al., 2019b)
necessary to improve the current representation of tropospheric $O_3$ in global chemical models
(Young et al., 2018). Spatial and temporal representativeness of $O_3$ observations is currently the
biggest source of uncertainty when inferring $O_3$ climatology in the free troposphere, even in
regions where observation are abundant but not ideally distributed (Lin et al., 2015b; Tarasick et
al., 2019b). Most studies reporting global $O_3$ distribution use satellite observations (Edwards et
al., 2003; Fishman et al., 1990, 1991; Thompson et al., 2017; Wespes et al., 2017; Ziemke et al.,
2005, 2006, 2017), modeling analyses (Hu et al., 2017), or observations spatially expanded using
back trajectory calculations (e.g., Liu et al., 2013; Tarasick et al., 2010). While useful, these
studies come with somewhat large uncertainties, as recently noted by reports from the
Tropospheric Ozone Assessment Report (TOAR), and thus require additional in situ observations
to be used as a validation bench-mark (Tarasick et al., 2019b; Young et al., 2018).

The Atmospheric Tomography mission (ATom, https://espo.nasa.gov/atom) was a NASA

Earth Venture airborne field project to address the sparseness of atmospheric observations over
remote ocean regions by systematically sampling the troposphere over the Pacific and Atlantic
basins along a global-scale circuit (Fig. 1). ATom deployed an extensive payload on the NASA
DC-8 aircraft, measuring a wide range of chemical, microphysical, and meteorological
parameters in repeated vertical profiles from 0.2 km to over 13 km altitude, from the Arctic to
the Antarctic over the Pacific and Atlantic Oceans, in four separate seasons from 2016 to 2018.
ATom built on a previous study, the HIAPER Pole-to-Pole Observations mission (HIPPO,
https://www.eol.ucar.edu/field_projects/hippo). The goal of HIPPO was to measure atmospheric
distributions of important greenhouse gases and reactive species over the Pacific Ocean, from the
surface to the tropopause, five times during different seasons from 2009 to 2011. Together,
ATom and HIPPO provide recent and comprehensive information about the altitudinal,
latitudinal, and seasonal composition of the remote troposphere over the Pacific, and over the
Atlantic for ATom. In addition, ATom and HIPPO sampling strategies were designed to deliver
an objective climatology of key species to enable modelling of air parcel reactivity of the remote
troposphere (Prather et al., 2017).

Here we use existing ozonesonde and commercial aircraft observations of $O_3$ at selected

locations along the ATom and HIPPO circuits to provide a climatological context for the
altitudinal, latitudinal, and seasonal distributions of $O_3$ derived from the systematic airborne in
situ "snapshots". Long-term $O_3$ observations are obtained from decades of ozonesonde vertical
profiles (e.g., Oltmans et al., 2013; Thompson et al., 2017) and from ~60,000 flights using the
In-service Aircraft for a Global Observing System (IAGOS) infrastructure (Petzold et al., 2015;
http://www.iagos.org). Ozonesondes have typically been launched weekly for two decades or
more, depending on the site, and have sampled a wide range of air masses across the globe, from
$O_3$-poor remote surface locations to the $O_3$-rich stratosphere. IAGOS commercial aircraft have
provided daily measurements in the upper troposphere and lower stratosphere (UTLS) for the
past 25 years, especially over the northern midlatitudes between America and Europe.
Combined, the ozonesonde and IAGOS datasets offer robust measurement-based climatologies
that quantify the full expected range of atmospheric $O_3$ variability with altitude and season.
The in-situ data from temporally-limited intensive field studies can be placed in context by
comparing them with long-term ozonesonde and commercial aircraft monitoring data. Evaluating
the representativeness of in situ observations from airborne campaigns by comparing them to
longer-term observational records is a critical exercise never before done at such a global scale.
We show that ATom and HIPPO measurements capture the spatial and, in some cases, temporal
dependence of $O_3$ in the remote atmosphere, thus highlighting the usefulness of airborne
observations to fill in the gaps of established but limited $O_3$ climatologies and other similarly
long-lived species. Then, we use the geographically extensive ATom and HIPPO vertical profile
data to establish a more complete measurement-based benchmark for $O_3$ abundance and
distribution in the remote marine atmosphere.

**2. Measurements**
2.1 ATom
The four ATom circuits occurred in July–August 2016 (ATom-1), January–February
2017 (ATom-2), September–October 2017 (ATom-3), and April–May 2018 (ATom-4), thus
spanning all four seasons in both hemispheres over a two-year timeframe (Table S1). The
mission in total consisted of 48 science flights and 548 vertical profiles distributed nearly equally
along the global circuit. All four deployments completed roughly the same loop, starting and
ending in Palmdale, California, USA (Fig. 1). A notable addition during ATom-3 and -4 were
out-and-back flights from Punta Arenas, Chile to sample the Antarctic troposphere and UTLS.
$O_3$ was measured using the National Oceanic and Atmospheric Administration (NOAA)
nitrogen oxides and ozone ($NO_yO_3$) instrument. The $O_3$ channel of the $NO_yO_3$ instrument is
based on the gas-phase chemiluminescence (CL) detection of ambient $O_3$ with pure NO added as
a reagent gas (Ridley et al., 1992; Stedman et al., 1972). Ambient air is continuously sampled
from a pressure-building ducted aircraft inlet into the $NO_yO_3$ instrument at a typical flow rate of
$1025.0 \pm 0.2$ standard cubic centimeters per minute (sccm) in flight. Pure NO reagent gas flow
delivered at $3.450 \pm 0.006$ sccm is mixed with sampled air in a pressure ($8.00 \pm 0.08$ Torr) and
temperature ($24.96 \pm 0.01$ °C) controlled reaction vessel. NO-induced CL is detected with a dry-
ice-cooled, red-sensitive photomultiplier tube and the amplified digitized signal recorded using
an 80 MHz counter; pulse coincidence corrections at high count rates were applied, but are
negligible for the data presented in this work. The instrument sensitivity for measuring $O_3$ under
these conditions is $3150 \pm 80$ counts per second per part per billion by volume (ppbv) averaged
over the entire ATom circuit. CL detector calibrations were routinely performed both on the
ground and during flight by standard addition of $O_3$ produced by irradiating ultrapure air with
185 nm UV light and independently measured using UV optical absorption at 254 nm. All $O_3$
measurements were taken at a temporal resolution of 10 Hz, averaged to 1 Hz, and corrected for
the dependence of instrument sensitivity on ambient water vapor content (Ridley et al., 1992).
Under these conditions the total estimated 1 Hz uncertainty at sea level is $\pm (0.015$ ppbv $+ 2$ %).
A commercial dual-beam photometer (2B Technologies model 211) based on UV optical
absorption at 254 nm also measured $O_3$ on ATom, with an estimated uncertainty of $\pm$ (1.5 ppbv +
1 %) at a 2-second sampling resolution. Comparison of the 2B absorption instrument $O_3$ data to
the $NO_yO_3$ CL instrument $O_3$ data agreed to within combined instrumental uncertainties, lending
additional confidence to the $NO_yO_3$ CL instrument calibration. For the ATom project we use
$NO_yO_3$ instrument $O_3$ data in the following analyses.
Data from two CO measurements were combined in this analysis. The Harvard quantum
cascade laser spectrometer (QCLS) instrument used a pulsed quantum cascade laser tuned at
$\sim$2160 cm$^{-1}$ to measure the absorption of CO through an astigmatic multi-pass sample cell with
76 m path length and detection using a liquid-nitrogen-cooled HgCdTe detector (Santoni et al.,
2014). In-flight calibrations were conducted with gases traceable to the NOAA World
Meteorological Organization (WMO) X2014A scale, and the QCLS observations have an
accuracy and precision of 3.5 and 0.15 ppb for 1 Hz data, respectively. CO was also measured by
the NOAA cavity ring-down spectrometer (CRDS, Picarro, Inc., model G2401-m; Karion et al.,
2013) in the 1.57 μm region with a total uncertainty of 5.0 ppbv for 1 Hz data. The NOAA
Picarro data were also reported on the Wolrd Meteorological Organization (WMO) X2014A
scale. The combined CO data (CO-X) used here corresponds to the QCLS data, with the Picarro
measurement used to fill calibration gaps in the QCLS time series.
Water ($H_2O$) vapor was measured using the NASA Langley Diode Laser Hygrometer
(DLH), an open-path infrared absorption spectrometer that uses a laser locked to a water vapor
absorption feature at $\sim$1.395 μm. Raw data are processed at the instrument's native $\sim$100 Hz
acquisition rate and averaged to 1 Hz with an overall measurement accuracy within 5 %.

2.2 HIPPO
The HIPPO mission consisted of five seasonal deployments over the Pacific basin
between 2009 and 2011, from the North Pole to the coastal waters of Antarctica (Wofsy, 2011).
HIPPO deployments consisted of two transects, southbound and northbound, and occurred in
January 2009 (HIPPO-1), October–November 2009 (HIPPO-2), March–April 2010 (HIPPO-3),
June–July 2011 (HIPPO-4) and August–September 2011 (HIPPO-5). The platform used was the
NSF Gulfstream V (GV) aircraft. More details can be found in Table S1.

A NOAA custom-built dual-beam photometer based on UV optical absorption at 254 nm

was used to measure $O_3$ (Proffitt and McLaughlin, 1983). The uncertainty of the 1 Hz $O_3$ data is
estimated to be $\pm$ (1 ppbv + 5 %) for 1 Hz data. A commercial dual-beam $O_3$ photometer (2B
Technologies model 205) based on UV optical absorption at 254 nm was also included in the
HIPPO payload. Comparison of the 2B $O_3$ data to the NOAA $O_3$ data showed general agreement
within combined instrument uncertainties on level flight legs. For the HIPPO project we use
NOAA $O_3$ data in the following analyses.

Data from two CO measurements were combined in this analysis. The QCLS instrument

was the same instrument as used during ATom and described in section 2.1. CO was also
measured by an Aero-Laser AL5002 instrument using vacuum UV resonance fluorescence (in
the 170–200 nm range) instrument with an uncertainty of $\pm$ (2 ppbv + 3 %) at a 2-second
sampling resolution. The combined CO data (CO-X) used here corresponds to the QCLS data,
with the Aero-Laser measurement used to fill calibration gaps in the QCLS time series.

2.3 IAGOS

IAGOS is a European Research Infrastructure that provides airborne in situ chemical,

aerosol, and meteorological measurements using commercial aircraft (Petzold et al., 2015). The
IAGOS Research Infrastructure includes data from both the CARIBIC (Civil Aircraft for the
Regular Investigation of the atmosphere Based on an Instrument Container; Brenninkmeijer et
al., 2007) and MOZAIC (Measurements of OZone and water vapor by Airbus In-service
airCraft; Marenco et al., 1998) programs, providing measurements from ~60,000 flights since
1994. We note the relative lack of IAGOS data over the Pacific compared to the Atlantic (shorter
temporal record, lower flight frequency, and much fewer flights with concomitant $O_3$ and CO
measurements), and therefore limited the comparison to the Atlantic. Because commercial
aircraft cruise altitudes over the ocean are predominantly between 9 and 12 km, the comparison
between ATom and IAGOS is further limited to the UTLS (Fig. 1). More details are shown in
Table S1.

Identical dual-beam UV absorption photometers measured $O_3$ aboard the IAGOS flights.

An instrument comparison demonstrated that the photometers (standard model 49, Thermo
Scientific, modified for aircraft use) showed good consistency in measuring $O_3$ following an
inter-comparison experiment (Nédélec et al., 2015). The associated uncertainty is $\pm$ (2 ppbv + 2
%) at a 4-second sampling resolution (Thouret et al., 1998).

CO measurements were made using infra-red absorption photometers (standard model 48

Trace Level, Thermo Scientific, modified for aircraft use) with an uncertainty of $\pm$ (5 ppbv + 5
%) at a 30-second sampling resolution (Nédélec et al., 2003, 2015).

2.4 Ozonesondes

Ozonesondes have measured the vertical distribution of $O_3$ in the atmosphere for decades,

and provide some of the longest tropospheric records that are commonly used to determine
regional $O_3$ trends (Gaudel et al., 2018; Leonard et al., 2017; Oltmans et al., 2001; Tarasick et
al., 2019a; Thompson et al., 2017). Ozonesonde launching sites are operated by the NOAA
ESRL Global Monitoring Division (GMD), NASA Goddard's Southern Hemisphere Additional
OZonesondes (SHADOZ) program, the New Zealand National Institute of Water & Atmospheric
Research (NIWA), the National Meteorological Center of Argentina (SNMA) in collaboration
with the Finnish Meteorological Institute (FMI), or Environment and Climate Change Canada. A
more detailed description of each ozonesonde site and corresponding dataset can be found in
Tables S1 and S2. All sites use electrochemical concentration cell (ECC) ozonesondes that rely
on the potassium iodide electrochemical detection of $O_3$, and which provide a vertical resolution
of about 100 m (Komhyr, 1969). The associated uncertainty is usually $\pm$ (5–10 %) (Tarasick et
al., 2019b; Thompson et al., 2019; Witte et al., 2018).

2.5 Data analysis

In this analysis, ATom flight tracks were divided into the Atlantic and Pacific basins, and

further subdivided into five regions within those basins: tropics, and northern and southern
middle- and high-latitudes. Vertical profiles presented graphically in this paper show $O_3$ median
values and the $25^{th}$ to $75^{th}$ percentile range within the 0–12 km tropospheric column sampled by
the DC-8 aircraft. These medians were obtained by averaging with equal weight the individual
profiles within each region over 1 km altitude bins.

HIPPO flight tracks are illustrated in Figure 1. The flight segments used for comparison

with ATom were binned into the same Pacific latitude and longitude bands as for ATom. HIPPO
vertical profile data are derived using the same methodology as for ATom.
All IAGOS flight tracks over the northern and tropical Atlantic are represented in Figure
1 in green. The latitude bands used to parse IAGOS data are consistent with the ones used for
ATom. The longitude bands are 50° W to 20° W in the tropics, 50° W to 10° W in the northern
midlatitudes, and 110° W to 10° W in the northern high-latitudes. Variations of the longitude
band widths do not significantly affect the $O_3$ distributions measured by IAGOS. Data from all
flights from 1994 to 2017 were included in the IAGOS dataset considered here, and were then
divided into two altitude bins (8–10 km and 10–12 km) in order to better understand the
influence of different $O_3$ sources (e.g., anthropogenic, stratospheric) on these two layers of the
atmosphere.
We compare the ozonesonde measurements to ATom and HIPPO aircraft data sampled
within 500 km of each ozonesonde launching site, since we expect a robust correlation in the free
troposphere within this distance (Liu et al., 2009). We used the surface coordinates of the
ozonesonde sites because the in-flight coordinates of ozonesondes are not available at all sites.
For comparison with ozonesonde long term records, we consider three regions of the
atmosphere: boundary layer (0–2 km), free troposphere (2–8 km), and UTLS (8–12 km). For
each layer, we compared monthly $O_3$ distributions from ozonesondes with the corresponding
seasonal $O_3$ distributions from aircraft measurements using the skill score ($S_{score}$) metric (Perkins
et al., 2007). The $S_{score}$ is calculated by summing the minimum probability of two normalized
distributions at each bin center, and therefore measures the overlapping area between two
probability distribution functions. If the distributions are identical, the skill score will equal 100
% (see Fig. S1 for further examples). Note the $S_{score}$ is positively correlated with the size of the
bin used to compare distributions. Here we chose a bin size of 5 ppbv, which is larger than the
combined precision of ATom, HIPPO, and IAGOS measurements, but small enough to separate
distinct air masses and their influence on $O_3$ distribution. Variables such as the distance to each
ozonesonde launching site (500 km in this study), the bin size of the $O_3$ distributions (5 ppbv in
this study), and the length of each ozonesonde record (full length in this study) can shift the
vertically-averaged $S_{score}$ value by up to 8 % (Table S3). We therefore treat this 8 % as a rough
estimate of the precision of the $S_{score}$ values presented here.
All three techniques (chemiluminescence, UV absorption, and ECC) used to measure $O_3$
for the datasets analyzed in this work have been shown to provide directly-comparable accurate
measurements with well-defined uncertainties (Tarasick et al., 2019b).

2.6 Back trajectory analysis

Analysis of back trajectories for air masses sampled during airborne missions is useful to

examine the air mass source regions and causes for $O_3$ variability over the Pacific and Atlantic
Oceans. We calculated ten-day back trajectories using the Traj3D model (Bowman, 1993;
Bowman and Carrie, 2002) and National Centers for Environmental Prediction (NCEP) global
forecast system (GFS) meteorology. Trajectories were initialized each minute along all of the
ATom flight tracks.

**3.  Comparison of ATom and HIPPO $O_3$ distributions to longer-term observational**

**records**

Here we use existing ozonesonde and IAGOS observations of $O_3$ at selected locations

along the ATom and HIPPO circuits to provide a climatological context for $O_3$ distributions
derived from the systematic airborne in situ "snapshots". We quantify how much of $O_3$
variability, occurring on timescales ranging from hours to decades, was captured by the
temporally-limited HIPPO and ATom missions.

3.1. Comparison to ozonesondes

ATom and HIPPO explored the fidelity with which airborne missions represent $O_3$

climatology in the remote troposphere. Here, we show that aircraft-measured median $O_3$ follows
the seasonal ozonesonde-measured median $O_3$ cycle at most of the sites studied here, and at
almost all altitudes – with a few exceptions (Figs. 2 and 3). Figure 2 plots the monthly median $O_3$
measurements from the tropical ozonesonde sites in three altitude bins, along with the median
values obtained from HIPPO and ATom measurements. Figure 3 plots the same for the
extratropical sites. Figure 4 correlates the median $O_3$ measured by aircraft in Figures 2 and 3
with those measured by ozonesondes. At the Eureka site, the winter and spring ATom
deployments recorded a significantly lower median $O_3$ compared to the corresponding
ozonesonde monthly median $O_3$ in the 0–2 km range (Fig. 3). Eureka is frequently subject to
springtime $O_3$ depletion events at the surface due to atmospheric bromine chemistry, which is
well recorded by the ozonesonde record (Fig. 3; Tarasick and Bottenheim, 2002). Sampling
during $O_3$ depletion events significantly lowered the ATom winter and springtime $O_3$
distributions near this site. In the 2–8 km range, there is a very good seasonal agreement between
ATom/HIPPO and the ozonesondes (Fig. 4b). Most seasonal differences are found above 8 km
(e.g., ATom in February at Trinidad Head and in May at Eureka; Fig. 3) and can be linked to the
occurrence – or absence – of stratospheric air sampling during ATom and HIPPO. In the absence
of stratospheric air mixing (< 8 km in Fig. 4), ATom/HIPPO successfully capture a large fraction
of $O_3$ climatology everywhere (Figs. 4b and 4c).

Figures 5 and 6 show vertical profiles of $O_3$ distributions by season at each ozonesonde

site, along with comparisons to HIPPO and ATom vertical profiles. Our analysis reveals that $O_3$
distributions derived from the ATom and HIPPO seasonal "snapshots" capture 30–71 % of the 1
km-vertically binned $O_3$ distribution established by long-term ozonesonde climatologies. For the
nine ozonesonde sites considered here, ATom and HIPPO captured on average 53 %, 54 %, and
38 % of the $O_3$ distribution in the 0–2 km, 2–8 km, and 8–12 km altitude bins, respectively.

Larger differences between ATom/HIPPO and the ozonesonde records in the UTLS (8–

12 km) can be ascribed to $O_3$ variability from stratospheric–tropospheric exchange, which is not
always captured by the ATom and HIPPO missions. This increased $O_3$ variability in the UTLS is
well-described by the long term ozonesonde records at Lauder, Trinidad Head, Eureka, Ushuaia,
and Marambio (Figs. 3 and 6). In these middle- and high-latitude locations in both hemispheres,
$O_3$ variability is especially pronounced during winter and spring, time periods favorable to more
frequent stratospheric air mixing (Greenslade et al., 2017; Lin et al., 2015a; Tarasick et al.,
2019a). Furthermore, the probability of sampling stratospheric air masses at ATom and HIPPO
ceiling altitude (12–14 km) increases with latitude, resulting in a lower $S_{score}$ between the
ATom/HIPPO and ozonesonde datasets at the extra-tropical sites than at the tropical sites (Figs.
S2a and S2b).

In the boundary layer (0–2 km) of the remote troposphere, $O_3$ variability is predominantly

impacted by loss mechanisms. Ozonesonde records show instances of $O_3$ mixing ratios lower
than 10 ppbv throughout the year in the boundary layer at the nine sites studied here (Figs. 2 and
3). The lowest $O_3$ mixing ratios are a result of (a) photochemical destruction over the oceans in
the tropics (Monks et al., 1998, 2000; Thompson et al., 1993), (b) $O_3$-destroying halogen
emissions in polar regions in springtime (e.g., Fan and Jacob, 1992), and (c) transport of $O_3$-poor
oceanic air over the midlatitude sites (e.g., Neuman et al., 2012).
ATom and HIPPO best describe the $O_3$ distribution in the free troposphere (2–8 km; Figs.
S2a and S2b). This suggests that airborne campaigns can capture global baseline $O_3$ values,
along with the long-range transport of $O_3$ pollution plumes often lofted to this altitude range and
responsible for $O_3$ variability.

While ATom consisted of one transect per ocean per season, HIPPO covered the Pacific
twice per seasonal deployment (southbound and northbound). The 1 km-binned $S_{score}$ is on
average higher when two combined seasonal HIPPO flights (southbound and northbound) were
available to compare to ozonesonde records, as opposed to when comparing $O_3$ profiles from
individual HIPPO transects with ozonesonde records (Fig. S2c). In addition, two seasonal flights
during HIPPO reduced the occurrence of low $S_{score}$ values. This $S_{score}$ decrease from flying only
one Pacific transect only during ATom was traded for the increase of vertical profiles over the
Atlantic Basin, which was not sampled during HIPPO. Future airborne missions with multiple
seasonal vertical profiles over large-scale regions would be ideal to better depict the full range of
tropospheric $O_3$ variability.

3.2. Comparison to IAGOS
IAGOS $O_3$ and CO observations in the northern Atlantic UTLS provide a measurement-
based climatology at commercial aircraft cruise altitudes for comparison to ATom. Simultaneous
measurements of $O_3$ and CO are of particular interest because CO provides a long-lived tracer of
continental emissions, which helps to differentiate $O_3$ sources (Cohen et al., 2018). We note that
while IAGOS measurements encompass hundreds of seasonal flights (depending on the region),
ATom sampled within each latitude band and season on one or two flights only (Fig. 1). Thus,
variability in the UT that occurred on timescales longer than a day was not captured by ATom.
Consequently, it is not surprising to see that ATom systematically under-sampled tropospheric
$O_3$ (and CO) variability compared to IAGOS at all latitudes in the northern Atlantic (Figs. 7 and
8). ATom captured on average 40 % of the $O_3$ variability measured by IAGOS in the Atlantic
UTLS (Fig. 7), on par with the $S_{score}$ of 38 % obtained when comparing ATom and HIPPO to
ozonesonde data (see section 3.1).

In the middle- and high-latitudes, the shapes of the $O_3$ vs. CO scatterplots from IAGOS

data demonstrate that distinct sources contribute to $O_3$ levels in the UTLS (Figs. 8a and 8b;
Gaudel et al., 2015). The high $O_3$ (>150 ppbv) – low CO (<100 ppbv) range corresponds to
intrusions of stratospheric air, which were mostly sampled in the spring season during ATom,
supporting previous observations of increased stratospheric air mixing during this season (Lin et
al., 2015a; Tarasick et al., 2019a). The low $O_3$ (<50 ppbv) – low CO (<100 ppbv) range
corresponds to the tropospheric baseline air, whereas the intermediate $O_3$ (50–120 ppbv) – high
CO (>100 ppbv) range generally represents the influence of air masses transported from
continental regions. During ATom, high $O_3$ and low CO in the middle- and high-latitude UTLS
were typical of stratospheric and baseline tropospheric air mixing.

$O_3$ measured during IAGOS rarely exceeds 150 ppbv in the northern tropical Atlantic

UTLS (Fig. 8c). This is expected because the tropical tropopause is typically situated between 13
and 17 km altitude and IAGOS flights typically cruise below 12 km. Therefore, instances of
stratospheric intrusions at IAGOS flight altitudes are limited. $O_3$ measured during ATom in the
tropical Atlantic above 8 km was generally positively correlated with CO, showing the
contribution of tropospheric $O_3$ production from continental sources reaching high altitudes.
Given this variability, the ATom data do not capture the extrema of UTLS $O_3$ variability in the
IAGOS measurements (Figs. 7 and 8). However, the most frequently measured $O_3$ and CO
values from ATom overlap with the most frequently measured $O_3$ and CO values from IAGOS
(contours in Fig. 8), suggesting that ATom captured the mode of the $O_3$ and CO distributions
from IAGOS in the northern Atlantic UTLS.

**4.   $O_3$ distributions in the remote troposphere from ATom and HIPPO**

We have established the fidelity of ATom and HIPPO $O_3$ data by comparison to

measurement-based climatologies of tropospheric $O_3$ from well-established ozonesonde and
commercial aircraft monitoring programs. In the following sections we exploit the systematic
nature of the ATom and HIPPO vertical profiles to provide a global-scale picture of tropospheric
$O_3$ distributions in the remote atmosphere. Figure 9 presents the altitudinal, latitudinal, and
seasonal distribution of tropospheric $O_3$ during ATom and HIPPO. Higher $O_3$ was measured
during ATom & HIPPO in the Northern Hemisphere (NH) than in the Southern Hemisphere
(SH), both in the Pacific and in the Atlantic. This distribution gradient has previously been
shown by global $O_3$ mapping from modeling, satellite, and ozonesonde analyses (e.g., Hu et al.,
2017; Liu et al., 2013). This finding holds true throughout the tropospheric column from 0 to 8
km, both in the middle- and high-latitudes (Fig. S3). In the midlatitudes below 8 km, median $O_3$
ranged between 25 and 45 ppbv in the SH, and between 35 and 65 ppbv in the NH. In the high
latitudes below 8 km, median $O_3$ ranged between 30 and 45 ppbv in the SH, and between 40 and
75 ppbv in the NH. Notable features in the global $O_3$ distribution are discussed in more detail in
the following sections. Figure 10 presents the vertically-resolved distribution of tropospheric $O_3$
from 0–12 km for the Atlantic (ATom in green) and for the pacific (ATom in pink, HIPPO in
blue). $S_{score}$ values resulting from the comparison of HIPPO and ATom Pacific distributions are
shown with blue diamonds, and from the comparison of ATom Atlantic and Pacific distributions
with pink squares. Figure 11 is derived from Figure 10 and gives the $S_{score}$ values against altitude
in the first panel, as well as the relative difference of median $O_3$ from 0 to 8 km in the second
panel.

4.1. Tropics
**Vertical distribution.** $O_3$ is at a minimum in the tropical marine boundary layer (MBL),
especially over the Pacific (Fig. 10a). The lowest measured $O_3$ in this region was 5.4 ppbv in
May during ATom, and 3.5 ppbv in January during HIPPO. The tropical MBL is a net $O_3$ sink
owing to very slow $O_3$ production rates – NO levels averaged $22 \pm 12$ pptv in the Pacific and
Atlantic MBL during ATom – and rapid photochemical destruction rates of $O_3$ in a sunny, humid
environment (Kley et al., 1996; Parrish et al., 2016; Thompson et al., 1993). Deep stratospheric
intrusions into the Pacific MBL were not observed in ATom or HIPPO, in contrast to reports
from previous studies (e.g., Cooper et al., 2005; Nath et al., 2016). In the tropics, marine
convection within the intertropical convergence zone (ITCZ) is associated with relatively low $O_3$
values throughout the tropospheric column, with median $O_3$ mixing ratios less than 25 ppbv
below 4 km altitude in the tropical Pacific (Fig. 10a; Oltmans et al., 2001). The relative
difference between ATom Atlantic and Pacific median $O_3$ in the tropics below 8 km is
consistently higher than a factor of 1.5, with an average $S_{score}$ of 43 % (Figs. 10a and 11b). We
ascribe this difference to $O_3$ production from biomass burning (BB) emissions in the continental
regions surrounding the tropical Atlantic; back trajectories from the ATom flight tracks show the
tropical Atlantic is strongly affected by transport from BB source regions in both Africa and
South America (Fig. S4; Jensen et al., 2012; Sauvage et al., 2006; Stauffer et al., 2018;
Thompson et al., 2000). In addition, the positive correlation of $O_3$ enhancements with black
carbon (Katich et al., 2018) and reactive nitrogen species (Thompson et al., personal
communication) also indicate BB influence. Although ATom and HIPPO data show evidence for
extensive and widespread BB influence on $O_3$ in the Pacific as well, $O_3$ mixing ratios are
consistently more elevated throughout the tropospheric column in the Atlantic. One reason is
closer proximity of the mid-ocean Atlantic flight tracks to $O_3$ precursor source regions. These
findings confirm studies that previously highlighted the impact of African BB emissions on $O_3$
production in the tropical Atlantic (e.g., Andreae et al., 1994; Fishman et al., 1996; Jourdain et
al., 2007; Williams et al., 2010). Lightning $NO_x$ also play a role in the buildup of $O_3$ over the
tropical Atlantic at certain times of year (Moxim and Levy, 2000; Pickering et al., 1996).

**Seasonality.** The seasonal variation of vertical profiles of $O_3$ in the tropics is lower

throughout the column compared to the extra-tropics (Fig. 12), in part due to less stratospheric
influence at the highest tropical altitudes. The remoteness of the tropical Pacific flight paths from
continental pollution sources also drives the lower seasonal variability here compared to the
tropical Atlantic, where BB influence peaks in June–August and October–November,
characterized by high $O_3$ (> 75 ppbv) and high CO (>100 ppbv) (Fig. 13f), significantly
increasing the $O_3$ vertical distribution compared to the other seasons (Figs. 12c, 12h, and 12m).
Finally, photochemistry, which regulates $O_3$ net balance in the troposphere, is less seasonally
variable in the tropics than in the extra-tropics, where the photolysis frequency of $O_3$ ($j(O_3)$) and
photochemical production of $O_3$ fluctuate annually with solar zenith angle.

**$O_3$ minima and maxima.** Coincident $O_3$ and CO enhancements were observed in the

tropical Atlantic for each ATom circuit (Figs. 9 and 13f), suggesting a year-round influence of
continental emissions and distinctive dynamics in this region (Krishnamurti et al., 1996;
Thompson et al., 1996). In the tropical Pacific, the April–May period stands out due to an $O_3$ and
CO enhancement episode during HIPPO (Fig. 9) that was attributed to the transport of
anthropogenic and BB emissions from southeast Asia (Shen et al., 2014). Deep convection in the
tropics brings $O_3$-poor (<15 ppbv) air to the upper troposphere (Kley et al., 1996; Pan et al.,
2015; Solomon et al., 2005). However, the spatial extent of these events remains poorly
constrained. Results from ATom and HIPPO suggest that deep convection can loft $O_3$-poor air at
least up to 12 km (the altitude ceiling of this study) in the tropical Pacific, and occurred more
frequently between January and May (Figs. 12c and h). During the rest of the year, $O_3$-poor air
was typically confined below 4 km. Conversely, $O_3$-poor air is confined to the first 2 km in the
tropical Atlantic (Fig. S5). Meteorological analysis of tropical ozonesondes shows that
subsidence of higher-$O_3$ air aloft over the Atlantic is one reason $O_3$-poor air is found only in the
boundary layer (Thompson et al., 2000, 2012).

4.2. Middle- and high-latitudes

**Vertical distribution.** In the middle- and high-latitudes, tropospheric $O_3$ was generally at
a minimum in the MBL and increased with altitude. Above 8 km, increasing $O_3$ with altitude
(Figs. 10b–e) and its persistent anticorrelation with CO (Fig. 13) points to stratospheric air
sampling as the cause for higher $O_3$ variability in the extra-tropical UTLS, especially at high
latitudes where the tropopause is lower and wave breaking of the polar jet streams can lead to
stratospheric intrusions. As a result, the $S_{score}$ decrease above 8 km, summarized in Figure 11a, is
ascribed to variability in the influence of stratospheric air. ATom detected little change in the $O_3$
distribution over the Pacific Ocean since HIPPO, with a $S_{score}$ averaging 74 % in the 0–8 km
range. The relative difference between median $O_3$ values from HIPPO and ATom in the Pacific
is generally lower than 20 % (Fig. 11b). Similarly, the relative difference between median $O_3$
mixing ratios between ATom Atlantic and Pacific below 8 km is consistently lower than 20 %,
with an average $S_{score}$ of 75 % (Fig. 11b). The southern high-latitudes are the only region where
the $S_{score}$ below 8 km occasionally fell below 60 % (Fig. 10e). However, a lower $S_{score}$ was
expected there as the Atlantic vertical profile is based on only two seasonal flights to Antarctica,
whereas there were four seasonal flights in the Pacific. Additionally, HIPPO was less spatially
extensive – resulting in fewer data points – in this latitude bin compared to ATom (Fig. 1), which
could explain the low $S_{score}$ values when comparing the two missions (Fig. 10e). Nevertheless,
the similar $O_3$ distribution in the extra-tropical free troposphere above the two oceans is
consistent with an $O_3$ lifetime sufficiently long for rapid zonal transport to smooth out variations
in baseline $O_3$ distribution in the remote troposphere, across a relatively wide range of longitudes
(Figs. 10b–e). The comparison of $O_3$ seasonal cycles at remote ozonesonde launching sites of the
northern midlatitudes yields similar results and further supports this conclusion (Logan, 1985;
Parrish et al., 2020). However, the similarity of the $O_3$ distribution in the extra-tropical free
troposphere above the Atlantic and Pacific is not always evident in satellite-, modelling-, or
ozonesonde-derived maps (Gaudel et al., 2018; Hu et al., 2017; Ziemke et al., 2017).
Additionally, studies of the spatial representativeness of tropospheric $O_3$ monitoring networks
have also concluded that tropospheric $O_3$ distributions varied significantly with longitude,
especially in the northern middle- and high-latitudes over continents (Liu et al., 2013; Tilmes et
al., 2012). In contrast, the ATom findings stem from $O_3$ measurements predominantly over the
oceans, which likely reveal a different picture of $O_3$ longitudinal distribution away from regional
precursor emissions.
**Seasonality.** The extra-tropical vertical profiles of $O_3$ vary seasonally during ATom and
HIPPO. The summer season in the middle- and high-latitudes was remarkable over both oceans
and hemispheres for the steep $O_3$ gradients in the tropospheric column (Fig. 12 in black). In the
MBL, median $O_3$ was consistently under 25 ppbv in the summer, whereas $O_3$ was over 25 ppbv
in other seasons. Low $O_3$ in the MBL in summer reflects the enhanced $O_3$ photochemical
destruction in this $NO_x$-limited region. Photochemical destruction decreases in dry air in the
upper troposphere, thus leading to the steep $O_3$ gradients observed here. The summer $O_3$
minimum was especially apparent in the high latitudes of the southern Pacific during ATom and
extended well above the MBL into the free troposphere (Fig. 12 in black). $O_3$ mixing ratios were
highest in the tropospheric column during springtime in both hemispheres, and over both oceans
(Fig. 12 in gold). A notable exception occurred during springtime in the high latitudes of the NH,
where several $O_3$ depletion events were sampled in the lower legs of the Arctic transit. During
these events, $O_3$ mixing ratios lower than 10 ppbv were measured, resulting in a lower 25[th]
percentile of $O_3$ distribution at the lowest altitude compared to the other seasons (Figs. 12e and
12o in gold). A tropospheric $O_3$ springtime maximum has often been reported in the NH (e.g.,
Monks, 2000) when meteorology favors efficient transport of $O_3$ and precursors from continental
air from North America and Eurasia (Owen et al., 2006; Zhang et al., 2017, 2008). Another
contributing factor is the increased frequency of stratospheric air mixing in spring that
significantly contributes to higher $O_3$ levels (Lin et al., 2015a; Tarasick et al., 2019a). Further,
the tropospheric $O_3$ springtime maximum in the SH is often attributed to BB emissions reaching
a peak (Fishman et al., 1991; Gaudel et al., 2018), but stratospheric air mixing also occurs (Diab
et al., 1996, 2004; Greenslade et al., 2017). Here, the $O_3$/CO relationship in spring shows that the
enhanced stratospheric mixing with tropospheric air during this season, both in the northern and
southern middle- and high-latitudes, contributes to the increase in column $O_3$ (Fig. 13). Fall and
winter seasons shared similar features in the middle- and high-latitudes: no strong $O_3$ gradient
was measured in the free troposphere, and $O_3$ values varied over similar ranges – about 40 ppbv
in the NH and about 30 ppbv in the SH – during the two seasons (Fig. 12 in red and blue).
**$O_3$ enhancements.** The linear increase of $O_3$ with CO >100 ppbv highlights the
contribution of natural and anthropogenic pollution plumes lofted from continental areas into the
remote troposphere. In the NH, these events occur almost year-round (Figs. 13b–c and 13g–h).
Higher CO enhancements in the Pacific (Figs. 13g–h) than in the Atlantic (Figs. 13b–c) have
been observed before and attributed to sampling bias (Clark et al., 2015). Here, our findings
suggest a year-round influence of continental emissions on the Pacific atmosphere despite its
remoteness. Modeled back trajectories show that most air masses sampled in the NH during
ATom were influenced by long-range transport of continental emissions from Asia, Africa, and
North America (Fig. S6). Previous studies have shown anthropogenic and BB emission outflow
from Asia significantly contributed to $O_3$ pollution events measured over the northern Pacific or
in California (e.g., Heald et al., 2003; Jaffe et al., 2004; Lin et al., 2017). Intercontinental
transport of anthropogenic emissions from Europe can also contribute to the Asian outflow of
anthropogenic pollution (e.g., Bey et al., 2001; Liu et al., 2002; Newell and Evans, 2000).
Finally, $O_3$ enhancements in the northern Atlantic were frequently observed and attributed to
midlatitude anthropogenic and boreal forest fire emissions (e.g., Honrath et al., 2004; Martín et
al., 2006; Trickl et al., 2003). In the SH, polluted air is encountered more often in spring and
summer over the Atlantic, but springtime CO is greater than in other seasons over the Pacific
(Figs. 13d–e and 13i–j). During spring, median $O_3$ above 50 ppbv was measured throughout the
free troposphere in the southern midlatitudes (Fig. 12). Several air masses intercepted during
these flights originated from regions that were intensively burning at the time, notably equatorial
and southern Africa, Australia, and southern South America, contributing to the observed
enhanced $O_3$ and CO (Fig. S4). Our results expand on previous observation-based, but more
spatially and temporally limited, studies that highlighted collocated enhancements of $O_3$ and CO
at remote locations to show in situ evidence of frequent, large-scale influence of continental
outflow on $O_3$ in the remote troposphere in both oceans, and at almost all latitudes.

**5. Conclusion**
We present tropospheric $O_3$ distributions measured over remote regions of the Pacific and
Atlantic Oceans during two airborne chemical sampling projects: the four deployments of ATom
(2016–2018) and the five deployments of HIPPO (2009–2011). The data highlight several
regional- and large-scale features of $O_3$ distributions, and provide insight into current $O_3$
distributions in remote regions. The main findings are as follows:
- ATom and HIPPO provide a unique perspective on vertically-resolved global baseline $O_3$
distributions over the Pacific and Atlantic basins, and expand upon spatially-limited $O_3$
climatologies from long-term datasets to highlight large-scale features necessary for
model output and satellite retrieval validation.

- ATom and HIPPO $O_3$ data are consistent – where they overlap – with measurement-
based climatologies of tropospheric $O_3$ from well-established ozonesonde and
commercial aircraft monitoring programs. ATom and HIPPO seasonal median $O_3$
correlated well with corresponding seasonal median $O_3$ from ozonesondes ($R^2 > 0.7$),
giving confidence in the accurate depiction of the emerging global $O_3$ climatology by
these diverse research activities. ATom and HIPPO captured 30–71 % of $O_3$ variability
measured by ozonesondes launched in the vicinity of the aircraft flight tracks, and had the
same mode of the $O_3$ distribution as determined by IAGOS in the northern Atlantic
UTLS. This representativeness evaluation on global scales highlights the usefulness of
airborne observations to fill in the gaps of established but limited $O_3$ climatologies.
Higher $O_3$ loading in the NH compared to the SH is consistent with the heterogeneous
distribution of $O_3$ precursor emissions around the globe, mostly concentrated in the NH, a
result consistent with previous modeling studies and satellite observations. ATom
Atlantic vs. Pacific comparison reveals a similar $O_3$ distribution in the free troposphere
up to ~8 km in the middle- and high-latitudes, but not in the tropics. Similar $O_3$
distributions across latitude bands have been suggested in the past, but these studies were
limited to the northern midlatitudes. Conversely, other satellite, modeling, and
observation-based studies indicated significant $O_3$ longitudinal gradients. Here, our
findings are consistent with zonal transport smoothing the baseline $O_3$ distribution
longitudinally from the Pacific to the Atlantic. In the tropics, median $O_3$ mixing ratios are
about twice as high in the Atlantic than in the Pacific, due to a well-documented mixture
of dynamical patterns interacting with the transport of continental air masses.

-   A comparison of seasonal $O_3$ vertical profiles did not reveal a marked seasonality in the

tropics, but instead highlighted the influence of specific events, most notably BB

emissions from Africa and South America, which have been extensively documented in

the literature. In the extra-tropics, the summer season was characterized by a steeper

tropospheric $O_3$ gradient driven by very low $O_3$ abundance in the MBL. Fall and winter

seasons generally led to near-constant $O_3$ mixing ratios from the surface to the upper

troposphere, while the highest $O_3$ abundance was recorded during the spring season when

more frequent and intense stratospheric intrusions and transport of air masses from

continental regions occur. ATom and HIPPO provide the first airborne in situ vertically-

resolved $O_3$ climatology covering both the Atlantic and Pacific Oceans in the NH and in

the SH. They confirm and extend the current understanding of $O_3$ variability in the

remote troposphere, built over several decades by airborne campaigns, monitoring

networks, and satellite observations.

-   Overall, this paper highlights the value of the ATom and HIPPO datasets, which cover

spatial scales commensurate with the grid resolution of current Earth system models, and

further, are useful as a priori estimates for improved retrievals of tropospheric $O_3$ from

satellite remote sensing platforms. In addition, ATom and HIPPO in situ measurements

help to establish the quantitative legacy of global pollution transport and chemistry

through the evaluation of key, covarying species – in this case $O_3$ and CO, and reveal the

607       year-round pervasive influence of continental outflow on $O_3$ enhancements in the remote

troposphere. ATom and HIPPO datasets should be critical for improving the scientific

community's understanding of $O_3$ production and loss processes, and the influence of

anthropogenic emissions on baseline $O_3$ in remote regions. They provide a timely

addition to the Tropospheric Ozone Assessment Report (TOAR) effort to characterize the

global-scale $O_3$ distribution, and address some of the measurement gaps identified

therein.


**Author Contribution**
SCW and TBR designed the research (ATom and HIPPO). The measurements were done by IB,
JP, CRT, TC, RC, BD, GWD, JWE, RSG, EJH, KM, FLM, CS, and TBR. BJJ, RK, RQ, RS,
DWT, AMT, and JCW provided the ozonesonde measurements. HC, AG, and VT provided the
IAGOS measurements. Back trajectory calculations were provided by ER and KCA. IB, JP,
CRT, KCA, RC, AG, EJH, KM, DDP, RQ, ER, DWT, AMT, VT, JCW, SCW, and TBR
contributed to the discussion and interpretation of the results. IB, JP, and TBR wrote the paper.

**Competing interests**
The authors declare no competing interest.

**Acknowledgments**
We thank the ATom leadership team, science team, and DC-8 pilots and crew for contributions
to the ATom measurements. ATom was funded in response to NASA ROSES-2013 NRA
NNH13ZDA001N-EVS2. The authors acknowledge support by the U.S. National Oceanic and
Atmospheric Administration (NOAA) Health of the Atmosphere and Atmospheric Chemistry,
Carbon Cycle, and Climate Programs. SHADOZ ozonesondes are supported by the Upper
Atmosphere Research Program of NASA. Ozonesoundings at Marambio have been supported by
the Finnish Antarctic research program (FINNARP). The IAGOS program acknowledges the
European Commission for its support of the MOZAIC project (1994-2003) the preparatory phase
of IAGOS (2005-2013) and IGAS (2013-2016); the partner institutions of the IAGOS Research
Infrastructure (FZJ, DLR, MPI, KIT in Germany, CNRS, Météo-France, Université Paul Sabatier
in France, and University of Manchester, UK); the French Atmospheric Data Center AERIS for
hosting the database; and the participating airlines (Lufthansa, Air France, China Airlines, Iberia,
Cathay Pacific, Hawaiian Airlines) for transporting the instrumentation free of charge. We thank
J. A. Neuman, H. Angot, and O. Cooper for helpful discussions and careful editing of this
manuscript.

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

1163

1164

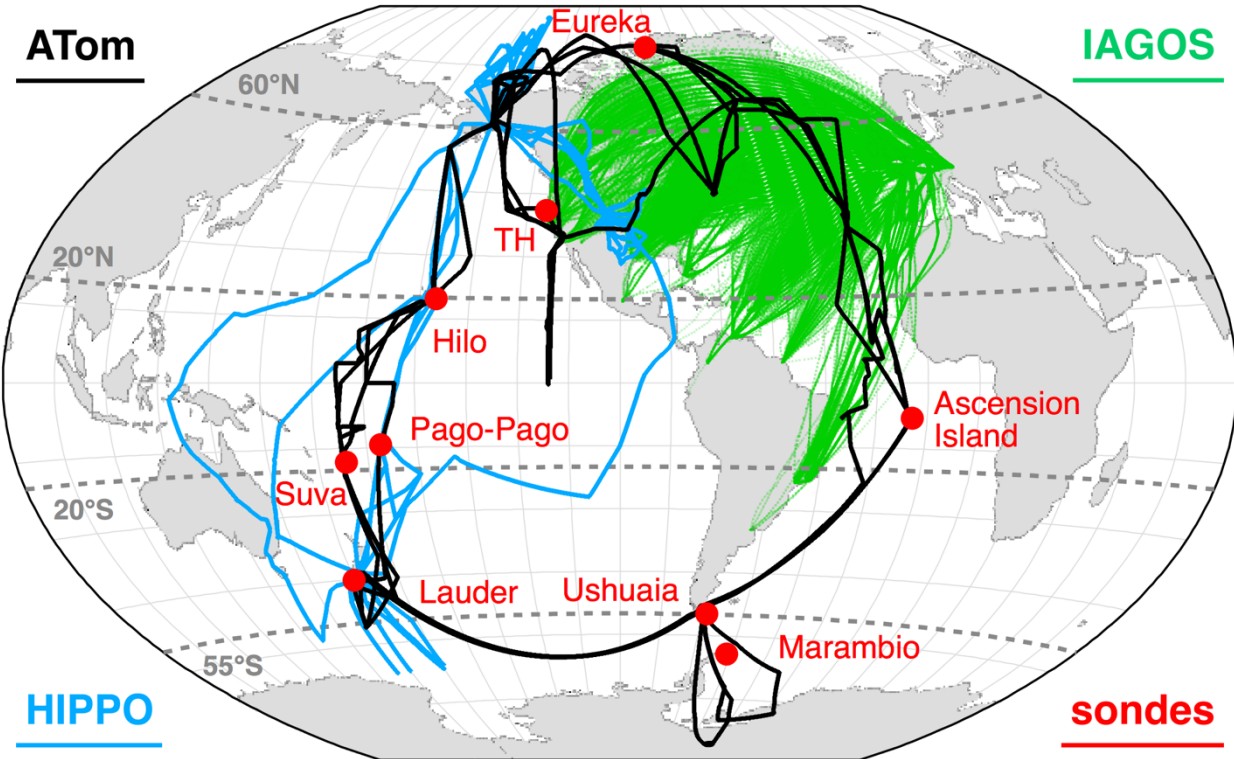

**Figure 1** The location and flight tracks of all $O_3$ monitoring platforms used in this work are illustrated with different markers and colors. The ATom flight track is in black, the HIPPO flight track is in blue, IAGOS flight paths are in green, and the ozonesonde launching sites are indicated by the red markers. The dotted grey lines define the latitudinal bands over which individual ATom and HIPPO profiles were averaged to derive a regional $O_3$ distribution: the tropics (20° S – 20° N), the midlatitudes (55° S – 20° S; 20° N – 60° N), and the high-latitudes (90° S – 55° S; 60° N – 90° N). Only data from remote oceanic flight segments of ATom and HIPPO missions were used in this work.

**Figure 2** Comparison of ATom (black squares) and HIPPO (blue diamonds) monthly median $O_3$ with ozonesonde (red circles) records from the four tropical sites. Markers indicate the median and the bars indicate the $25^{th}$ and $75^{th}$ percentiles. The three rows, from bottom to top, correspond to the boundary layer (0–2 km), the free troposphere (2–8 km), and the UTLS (8–12 km). The pink dots show every $O_3$ data point measured by ozonesondes for the timeframes indicated in Table S2.

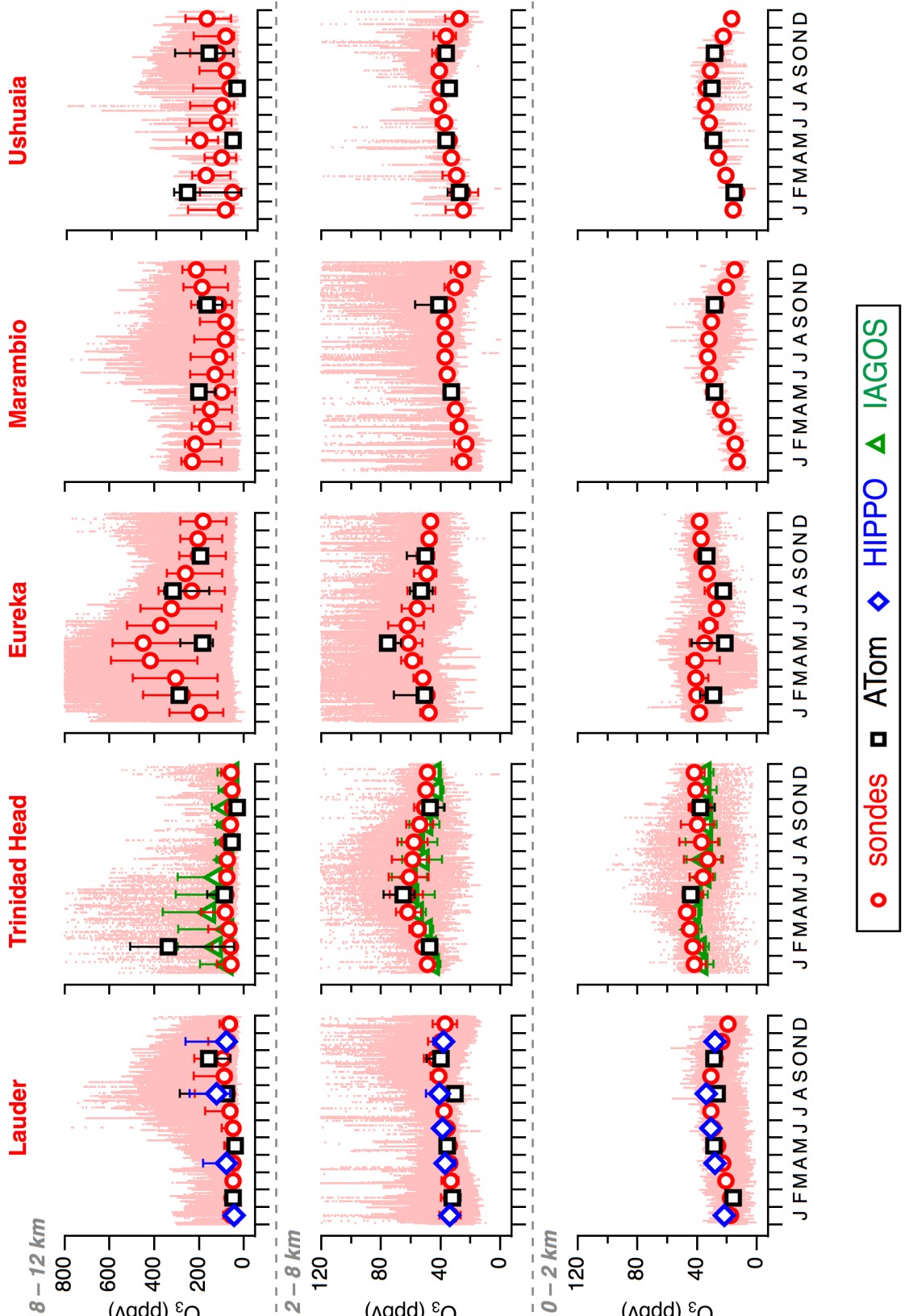

**Figure 3** Same as in Figure 2 but for ozonesonde launching sites located in the middle- and high-latitudes. $O_3$ data obtained from the IAGOS program (green triangles) during descents into San Francisco Bay-area airports were also added to the Trinidad Head site for comparison.

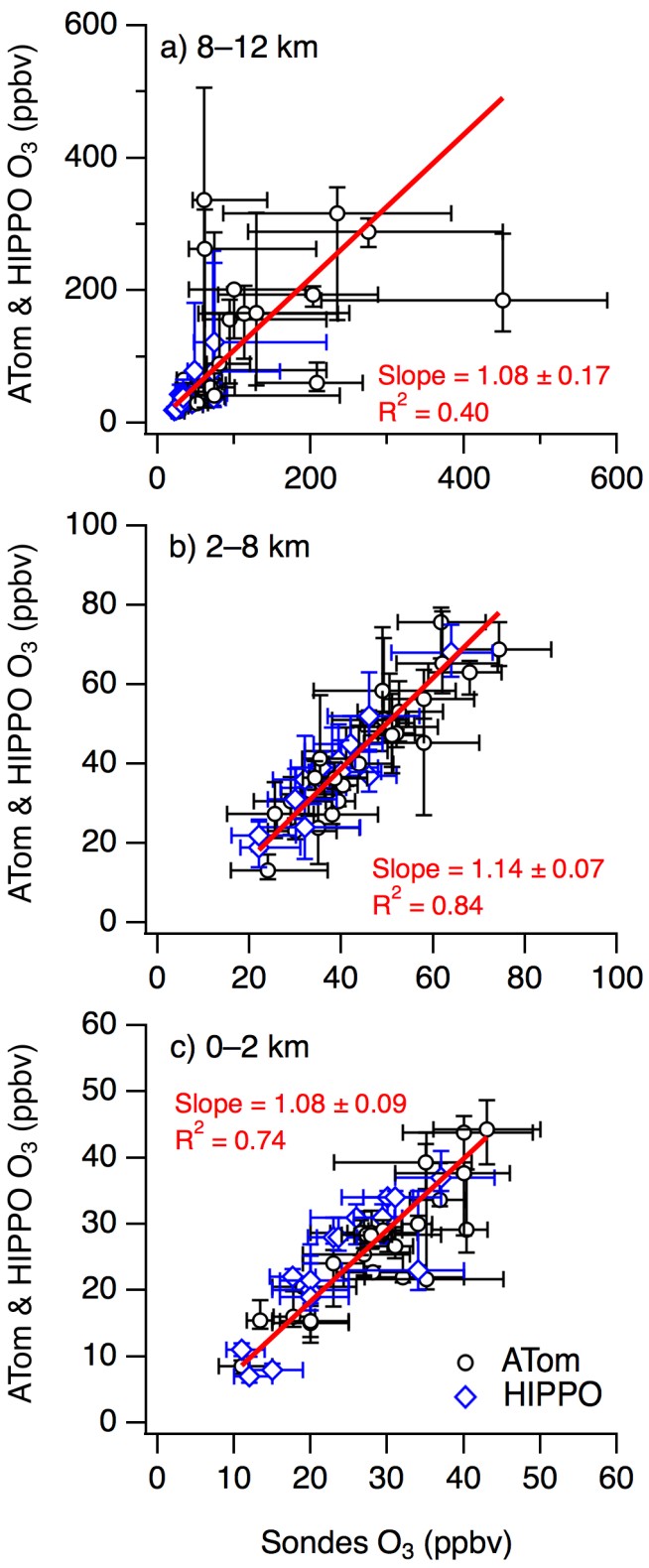

**Figure 4** ATom (black circles) and HIPPO (blue diamonds) combined monthly median O₃ vs. monthly median O₃ from ozonesondes at the nine sites considered in this study. The three panels

43–52

indicate the correlations for a) the UTLS (8–12 km), b) the free troposphere (2–8 km), and c) the boundary layer (0–2 km). The orthogonal regression fits are two-sided but not weighted.

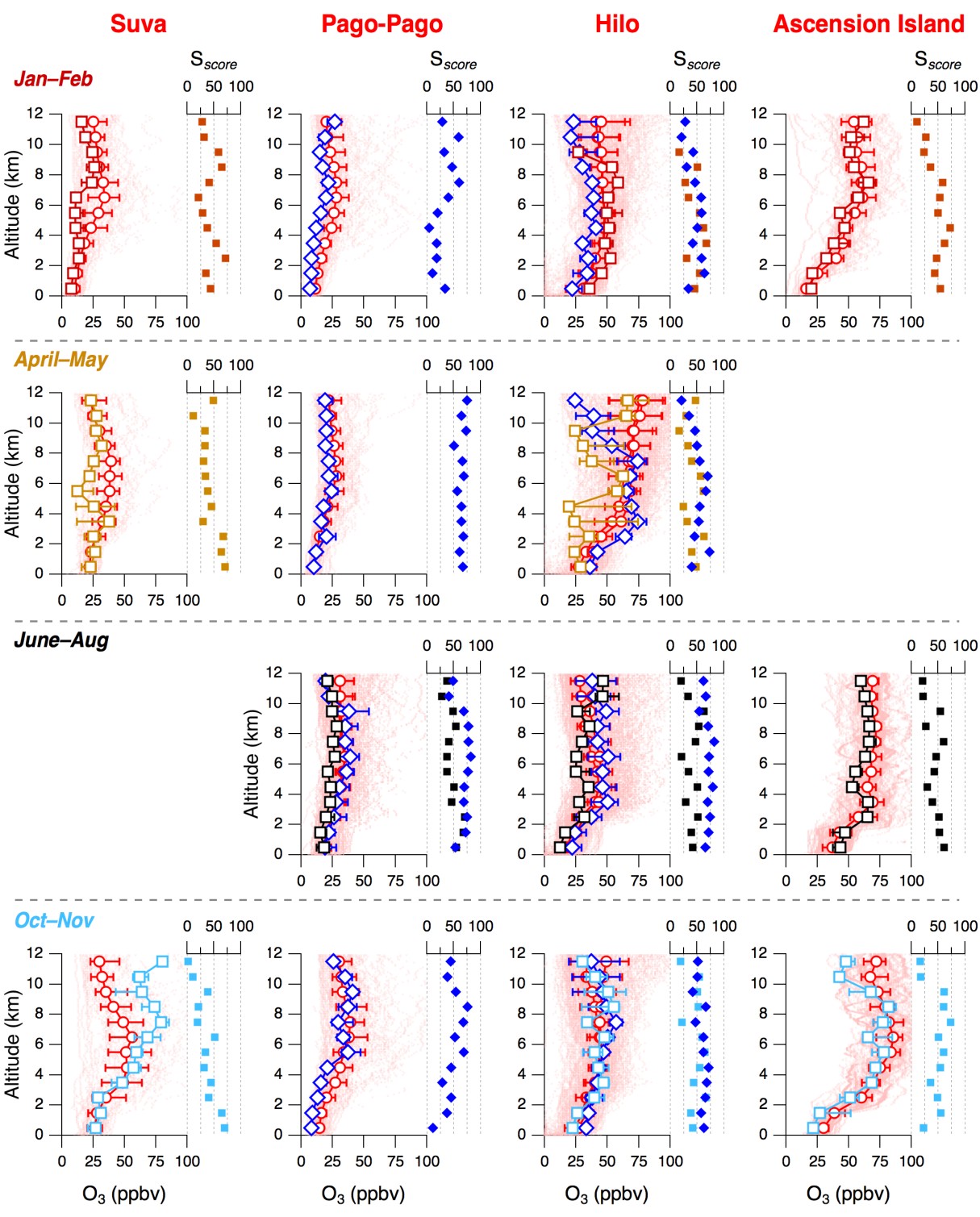

**Figure 5** Seasonal comparison of 1 km-vertically-binned ATom (colored squares) and HIPPO (blue diamonds) median $O_3$ with ozonesonde (red circles) records at four sites in the tropics (Suva in Fiji, Pago-Pago in American Samoa, Hilo in Hawaii, and Ascension Island). Markers indicate the median and the bars are the 25th and 75th percentiles. The $S_{score}$ is a metric of how well ATom

and HIPPO 1 km-binned $O_3$ probability distribution functions (PDFs) overlap with the corresponding 1 km-binned $O_3$ PDFs from ozonesondes. The $S_{score}$ shown with squares compares ATom with ozonesondes, and the $S_{score}$ shown with blue diamonds compares HIPPO with ozonesondes. The pink dots show every $O_3$ data point measured by ozonesondes for the timeframes indicated in Table S2.

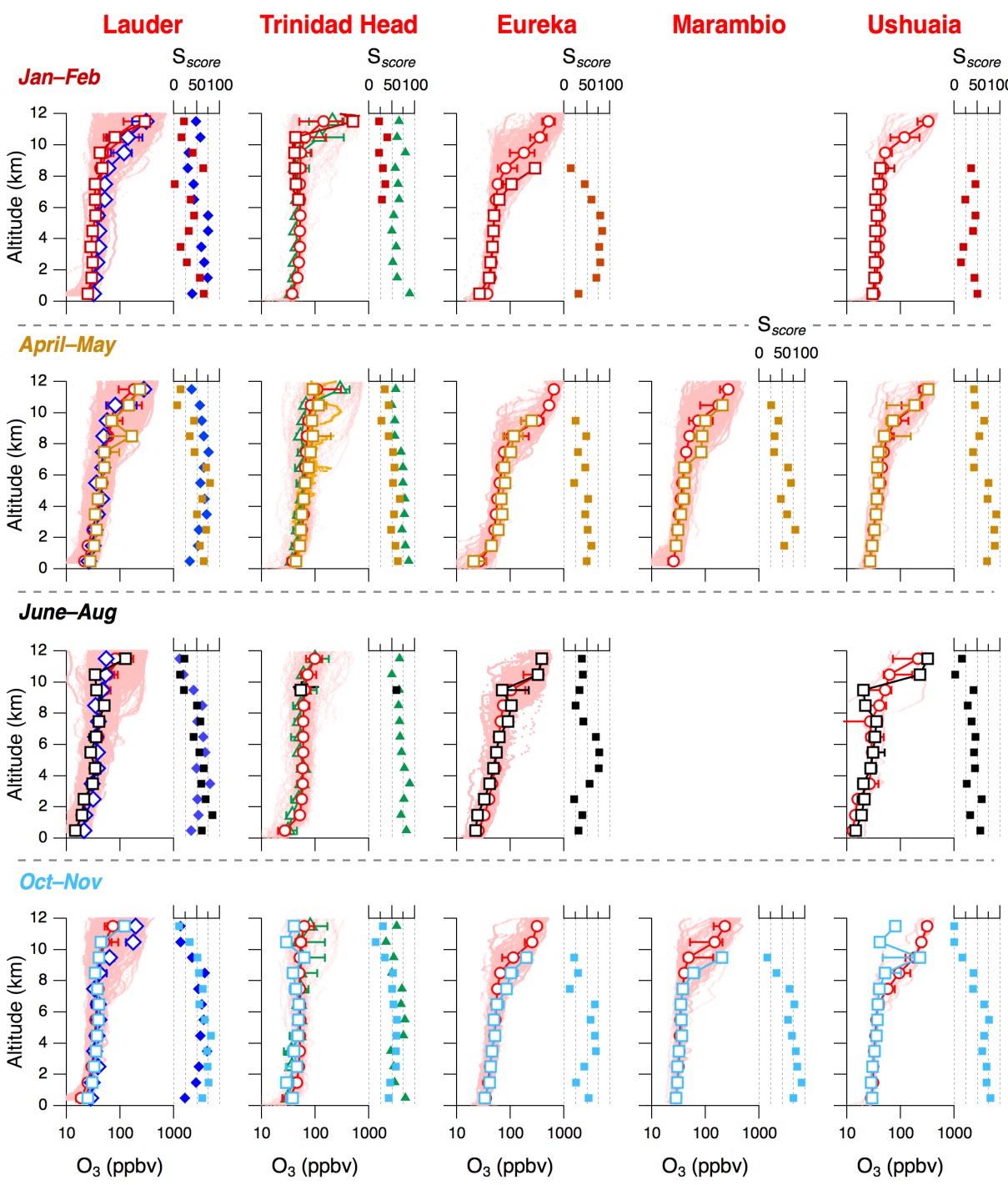

**Figure 6** Same as in Figure 5 but for ozonesonde launching sites located in middle- and high-latitudes (Lauder in New Zealand, Trinidad Head in the USA, Eureka in Canada, Ushuaia in Argentina, and Marambio in Antarctica). $O_3$ data obtained from the IAGOS program (green triangles) during descents into San Francisco Bay-area nearby airports were also added to the Trinidad Head site for comparison.

47–52

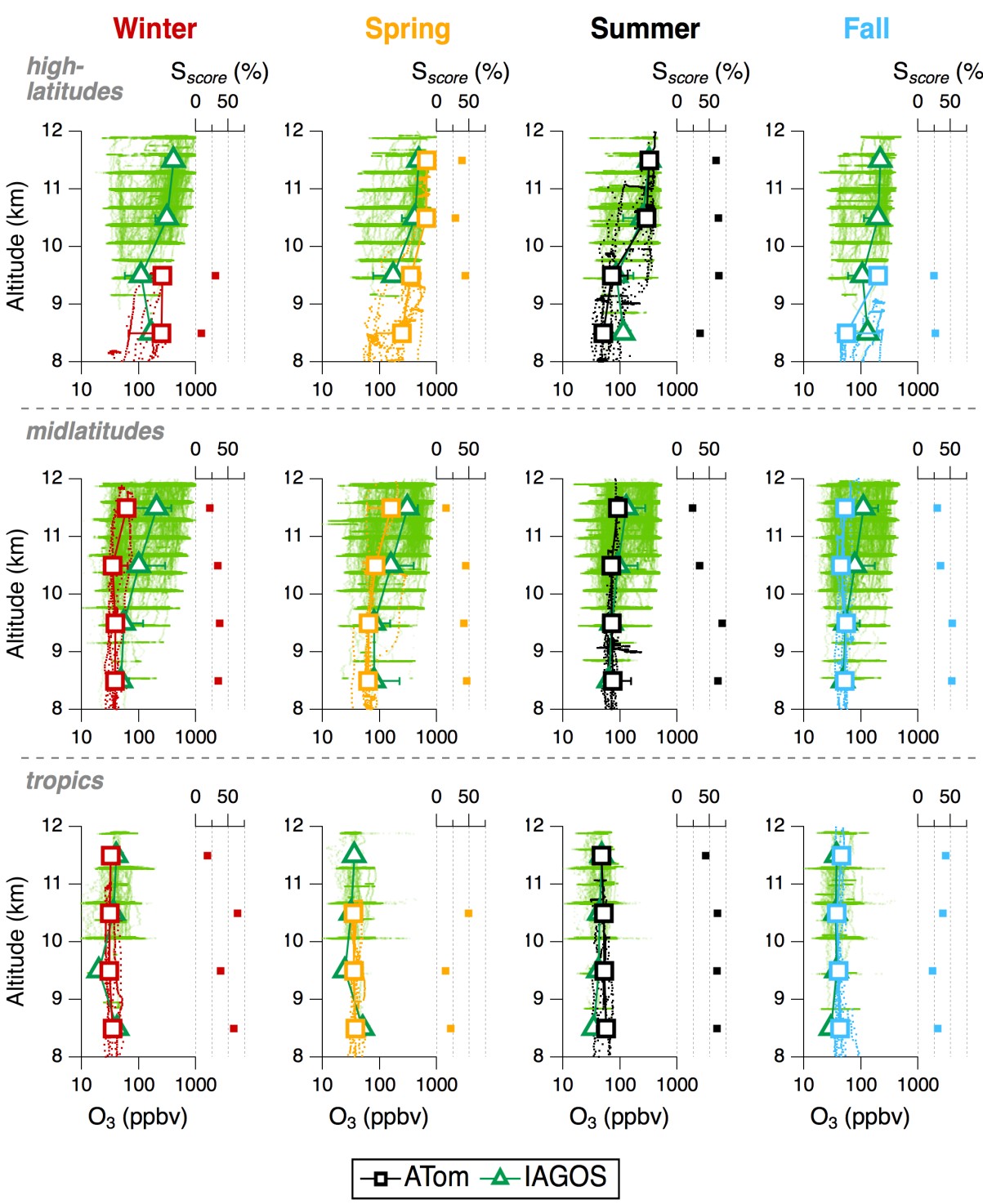

**Figure 7** Seasonal comparison of 1 km-binned ATom (colored squares) median $O_3$ with IAGOS (green triangles) in the northern Atlantic UTLS. Markers indicate the median and the bars are the 25th and 75th percentiles. The three different rows indicate the latitudinal bands. The four columns

indicate the seasons. The green dots show every $O_3$ data point measured by IAGOS flights for the timeframe indicated in Table S1.

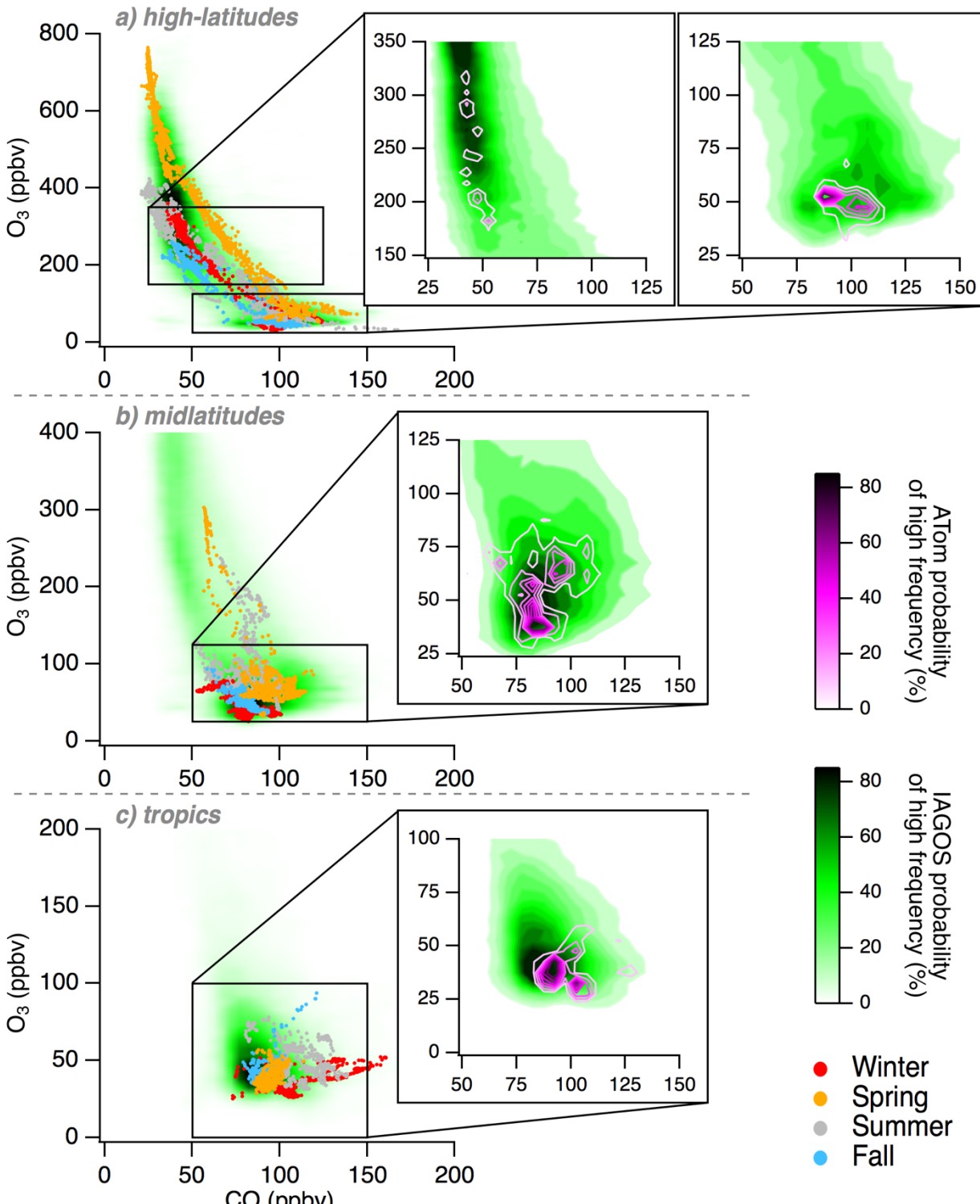

**Figure 8** IAGOS and ATom seasonal $O_3$ vs. CO scatterplots, with insets showing the most frequent $O_3$ values measured during IAGOS and ATom. ATom seasonal deployments are

colored according to the legend. The frequency gradient of $O_3$ counts is illustrated by the color scales (green for IAGOS, magenta for ATom). ATom measurements have been combined for the frequency gradients shown in the insets. The probability of high frequency refers to the probability of finding frequently measured $O_3$ values within the contour boundaries

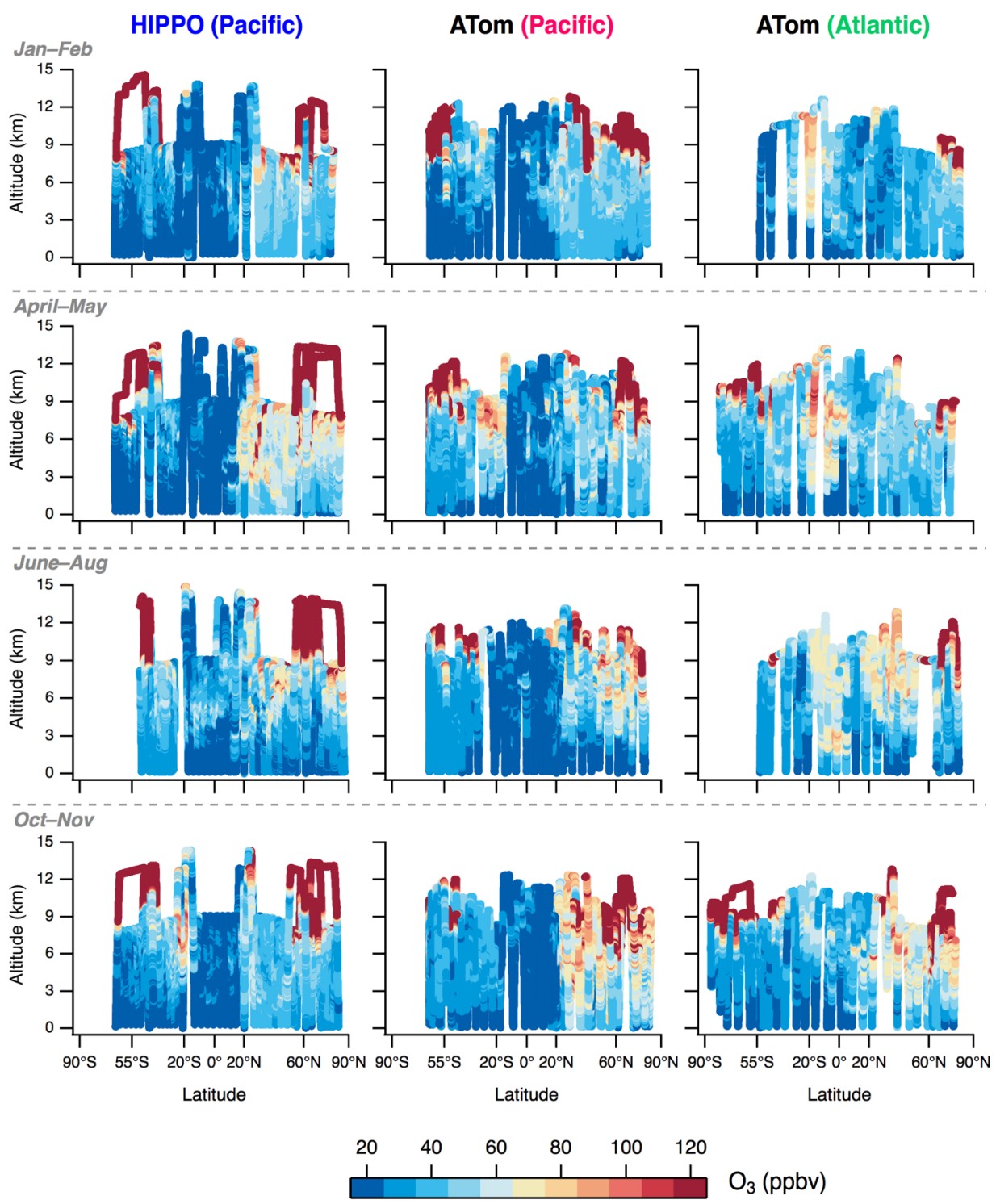

**Figure 9** Global-scale distribution of tropospheric O₃ for each ATom and HIPPO seasonal deployment. The rows separate the seasonal deployments, while the columns indicate the mission and the ocean basin. The O₃ color-scale ranges from 20 to 120 ppbv, and all values outside of this

range are shown with the same extremum color (red for values > 120 ppbv, blue for values < 20 ppbv). HIPPO deployments in June and August were combined together.

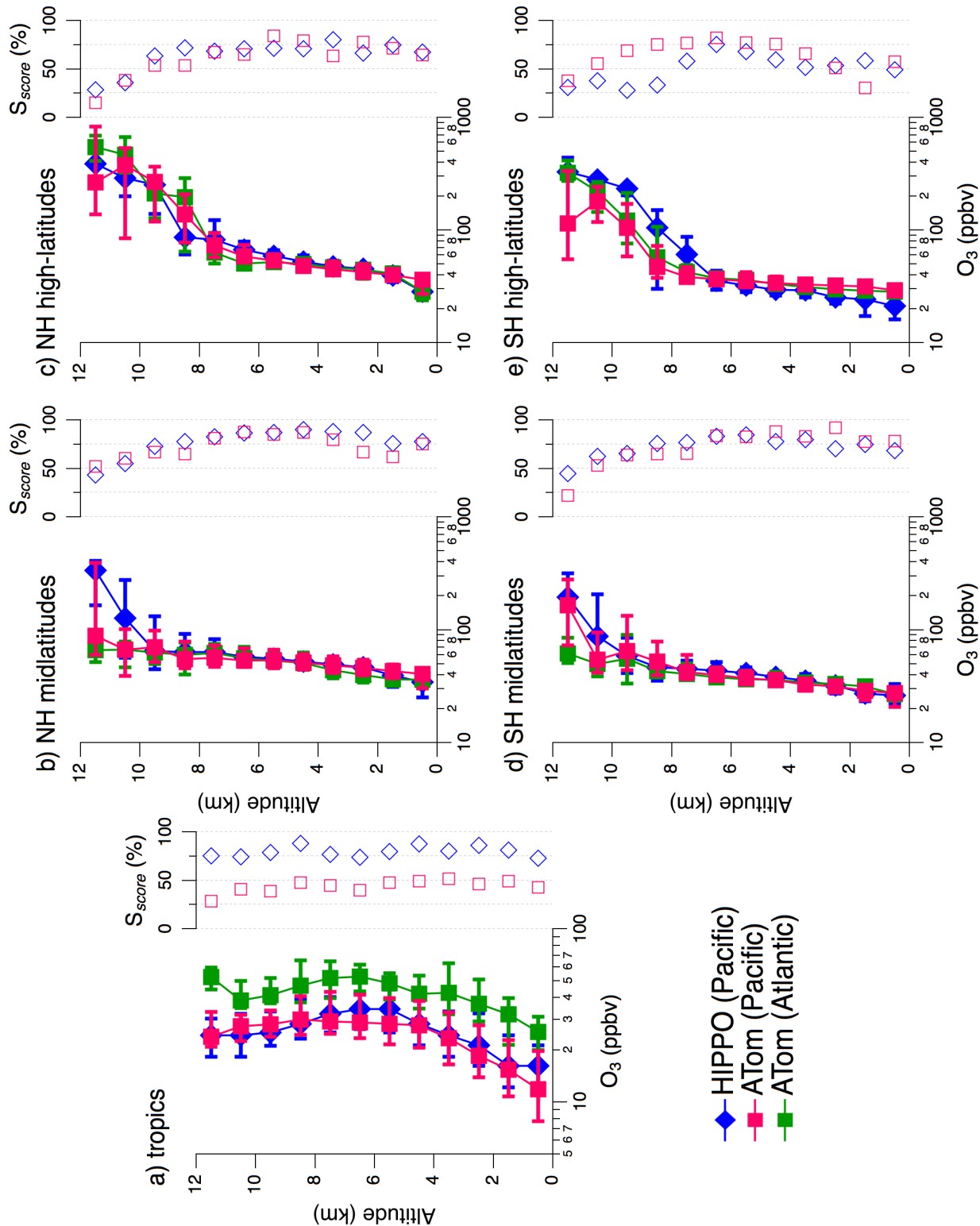

**Figure 10** Vertically-resolved O₃ distributions from 0–12 km are plotted for the Atlantic (ATom in green) and for the Pacific (ATom in pink, HIPPO in blue). The five broad latitude regions correspond to the data parsing illustrated by Fig. 1. Markers indicate median O₃, and bars are the

$25^{th}$ and $75^{th}$ percentiles, per 1 km altitude bin. Note the log scale on the x-axis. $S_{score}$ values resulting from the comparison of HIPPO and ATom Pacific distributions are shown with blue diamonds, and from the comparison of ATom Atlantic and Pacific distributions with pink squares.

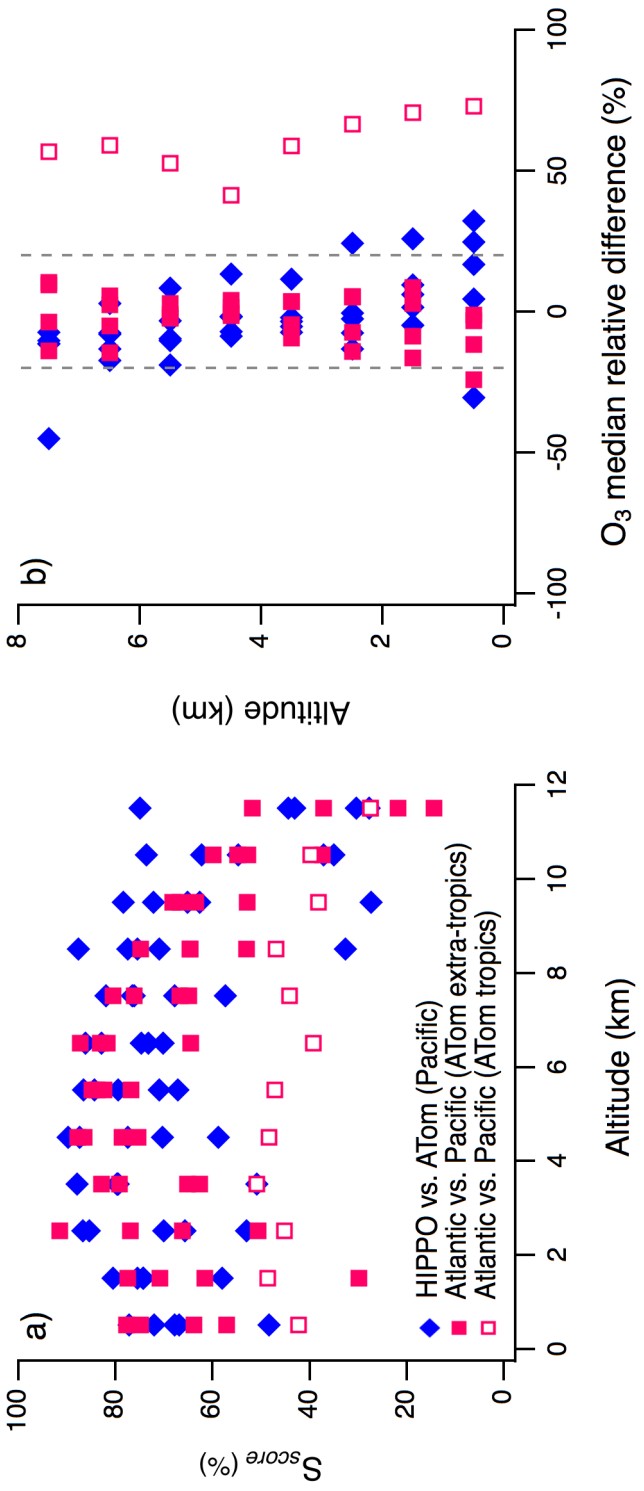

**Figure 11** All S$_{score}$ values from Fig. 10 are shown in panel a) and plotted against altitude. The HIPPO and ATom comparison in the Pacific basin is shown with blue diamonds, and a comparison of the Atlantic and Pacific basins during ATom is shown with filled pink squares for the extra-

tropics and open pink squares for the tropics. The relative difference of median $O_3$ from 0 to 8 km given in Fig. 10 is shown in panel b), with the same color and marker code as in panel a). The dotted grey lines indicate a relative difference of 20 %.

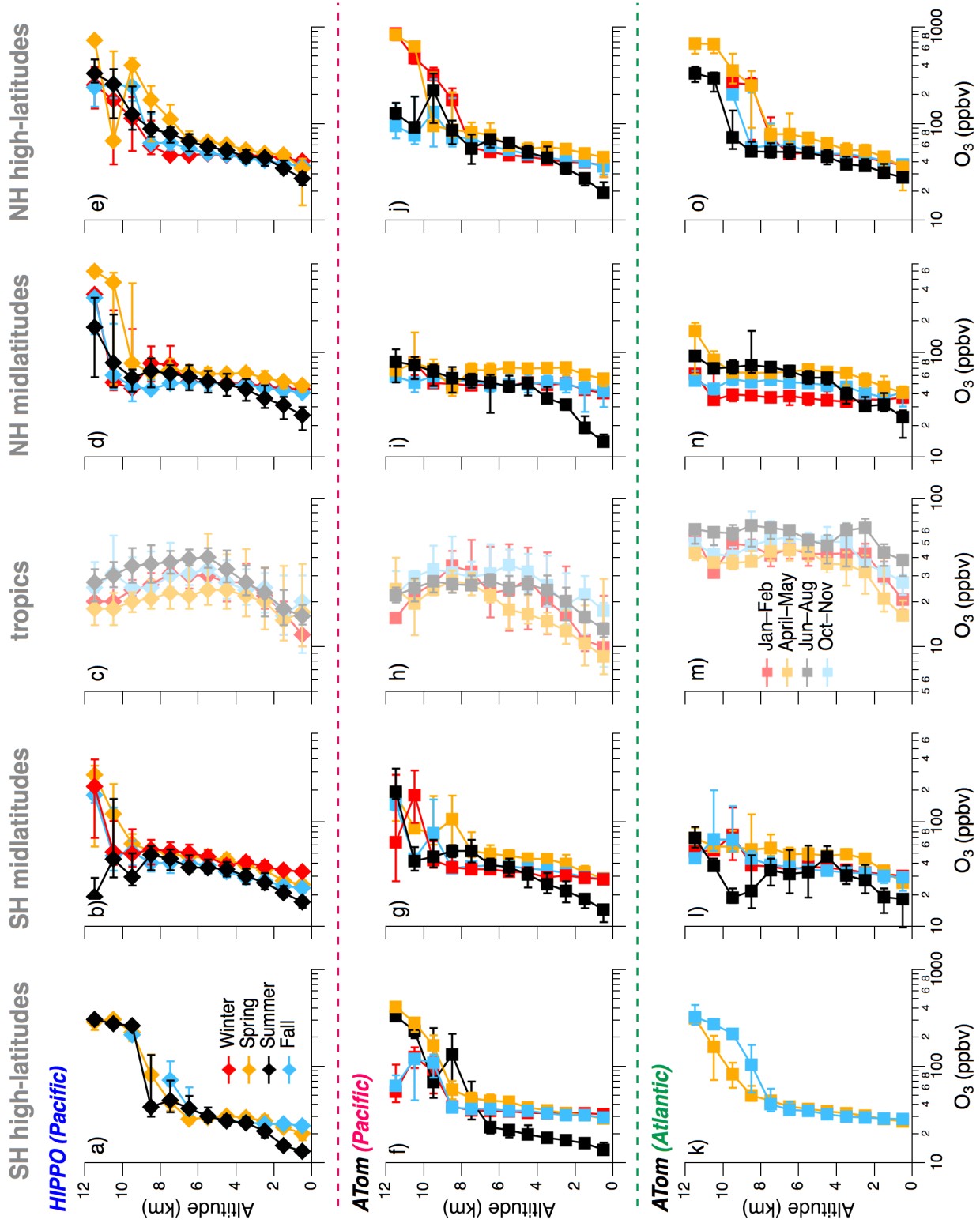

**Figure 12** Seasonal variability of regional O₃ distribution in the Pacific (HIPPO in the first and ATom in the second row) and in the Atlantic (ATom in the third row). The colors designate the

local seasons with red as winter, gold as spring, black as summer, and blue as fall (corresponding months are indicated for the tropics, with lighter colors). The markers and associated bars correspond to the median, 25th and 75th percentiles, respectively, of $O_3$ distribution in every 1 km altitude bin. Note the logarithmic scale on the x-axes in all panels, and the changing scale with latitudinal bin.

**Figure 13** O$_3$ vs. CO plots using combined ATom and HIPPO data. Each panel denotes a different latitudinal band in each basin. Seasonal deployments are colored according to the legend. Note the logarithmic scale on the y-axes in all panels, and the changing scale with latitudinal bin

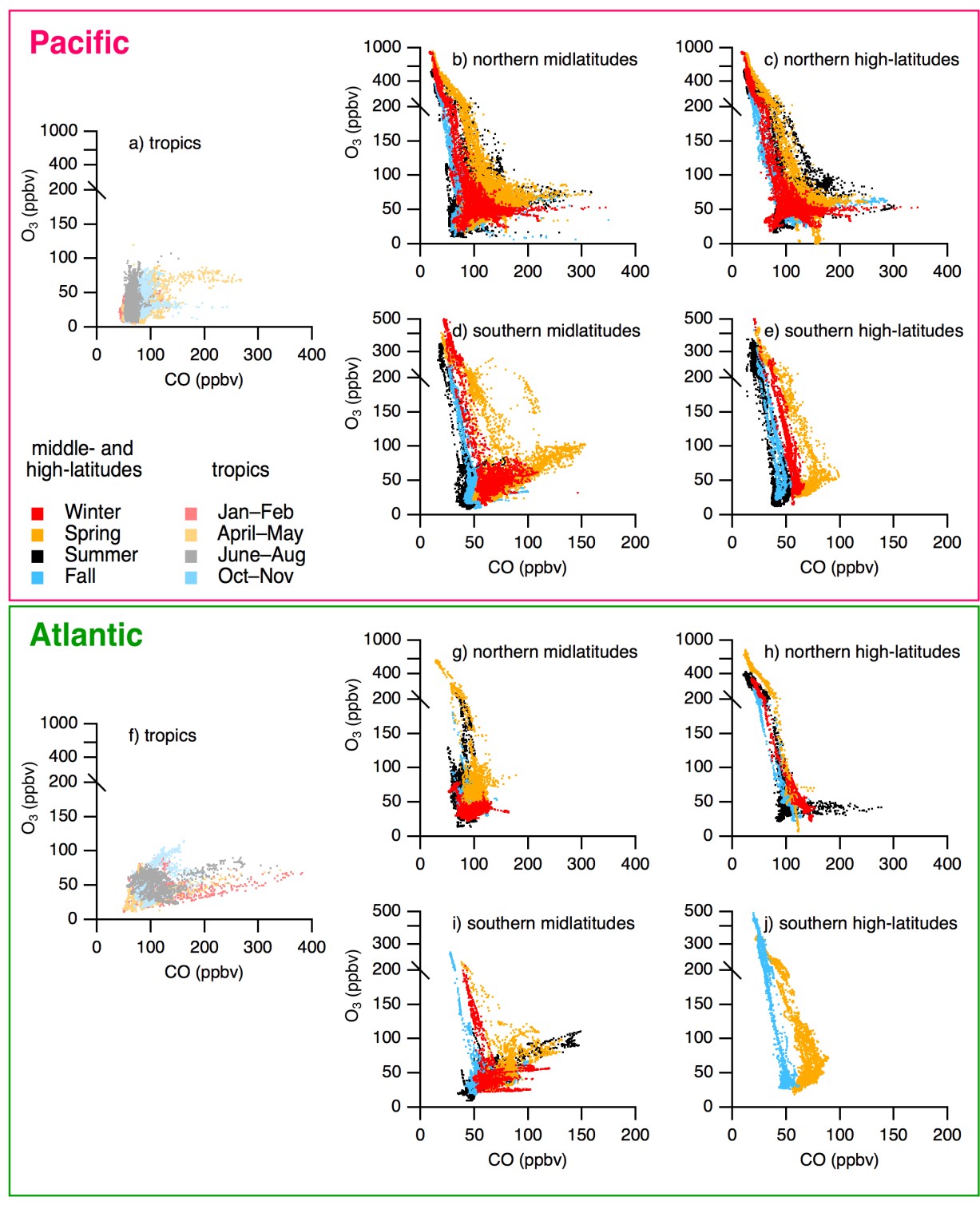