# Peer review of "Global-scale distribution of ozone in the remote troposphere from ATom and HIPPO airborne field missions."

_Atmospheric Chemistry and Physics, 2020_

## Referee Comment (RC1) · Anonymous Referee #1 · 7 May 2020

The paper presented by Ilann Bourgeois and a large number of co-authors is not bringing a large number of new knowledge, but it is presents a large number of observations of ozone over the Atlantic and the Pacific oceans during the 4 seasons. In this regard, the paper is very important because it helps getting a global view of the ozone distribution in regions that are not too close to sources of precursors. The paper makes use of important field campaigns including ATom and HIPPO. What is really nice is that the data from these airborne campaigns are compared with ozone sonde measurements (when possible) and data from IAGOS in the North Atlantic (civil aircraft measurements).

[Figure]

I have no major comments to make except to congratulate the teams involved for the wonderful airborne missions conducted under the scientific leadership of Steve Wofsy. The paper is very clearly written with appropriate references and justifications of the statements that are made. The methodology is clearly stated. The results are well presented and the discussion is clear.

I would have wished to see a bit more background material in the introduction, and maybe this can be added. What did we know about ozone above the two oceans from previous experimental studies including space observations and ozone sondes? What did these studies show regarding ozone in the tropics and in the extra-tropics as a function of season? What did we know about the influence above the oceans of African and American biomass burning and lightning? What about the plumes from industrialized countries? And at the end of the paper, does the study confirm what was known or its there any new findings that would change our understanding of the processes involved? Perhaps a few sentences on these issues in the introduction and the conclusions would make the paper more attractive.

---

## Referee Comment (RC2) · Anonymous Referee #2 · 11 May 2020

This paper reports tropospheric ozone distributions measured in ATom and HIPPO, compares them to ozonesondes and IAGOS to determine the consistency between the air sampled by ATom/HIPPO and long-term climatology, and comments on different features of the ozone distributions including vertical and seasonal distributions in the tropics and extratropics, and differences between the Pacific and Atlantic.

Ozone curtain data from ATom and HIPPO are worth reporting in the literature, and it is also of interest to place ATom and HIPPO in a broader climatological context, as can be done with ozone data but has broader implications for the ensemble of ATom/HIPPO measurements for which such long-term records don't exist. This is a useful though

limited contribution. However, a big problem with the current paper is that it claims to discover what are very well known features of the global ozone distribution, and it fails to credit previous work (notably from Jennifer Logan, Ed Browell's group, and many model papers) that made exactly the same points 1-3 decades ago. GTE campaign papers from Ed Browell's group using DIAL (PEM series, INTEX-B, SONEX, TRACE-A, TRACE-P...) report very similar ozone curtains as ATom and HIPPO. All the findings reported here in the Abstract and Conclusions can be found in the previous literature. Once this is corrected and previous literature is properly credited, then it is not clear what is actually new in this paper in terms of scientific findings. It represents a limited contribution, and whether this is worth publishing in ACP is an Editor's decision.

Specific comments:

1. Abstract, lines 20-22: it is well known that there is more ozone in NH than SH.

2. Abstract, lines 22-23: uniformity of free tropospheric ozone at northern mid-latitudes has been known since Logan JGR 1985 from ozonesonde data.

3. Abstract, line 24: the higher ozone over tropical Atlantic than Pacific is well known – there is extensive literature on this from SHADOZ, TRACE-A, satellite retrievals (Jerry Ziemke. . .)

4. Abstract, lines 26-28: Continental influences over the NH oceans have been known for decades (PEM-Tropics B, INTEX-B over Pacific come to mind), vertical structure is a well known feature from ozonesonde and GTE data and again this has been extensively discussed in literature including countless model papers.

5. Abstract: that last sentence is gratuitous.

6. Introduction: there has to be some serious review of previous knowledge on the tropospheric ozone distribution from ozonesondes, GTE aircraft campaigns, satellites, models. The authors are addressing a very old problem, on which there is a lot of literature. Ignoring that literature is not right, particularly because it leads to claims in

the Abstract and Conclusions that suggest that the authors are unaware of it.

7. Figures: there are way too many figures, and too much information on the figures – the eyes glaze, there's a lot of repetitiveness. It's hard to care about Sscore.

8. Line 306: "airborne campaigns can capture global baseline O3 values". Didn't we already know this from the GTE campaigns?

9. Lines 364-366, also 399-400, 409-411: we do know that ozone can be >100 ppb in biomass burning plumes over the South Pacific in spring (several papers coming out of PEM-Tropics A in Sep-Oct1996), seems like ATom/HIPPO just didn't hit them.

10. Lines 452-453 – uniformity of free tropospheric ozone at northern mid-latitudes has been known since Logan 1985 and has been shown in countless models.

11. Lines 496-497: continental influence on ozone over the North Pacific is hardly new.

12. Lines 516-, Conclusions: the conclusions list the same unwarranted claims of "valuable new insight into O3 distributions in remote regions" (lines 520-521) as the Abstract. All of the features presented here about the global ozone distribution have been amply documented in the literature.

---

## Author Comment (AC1) · 20 Jun 2020

Our responses are interspersed with the reviewers' comments (in bold font) and are given in normal black text (changes to the manuscript are shown in red). The line numbers where those changes appear in the revised paper are also given at that point.

**Reviewer 1**
**The paper presented by Ilann Bourgeois and a large number of co-authors is not bringing a large number of new knowledge, but it is presents a large number of observations of ozone over the Atlantic and the Pacific oceans during the 4 seasons. In this regard, the paper is very important because it helps getting a global view of the ozone distribution in regions that are not too close to sources of precursors. The paper makes use of important field campaigns including ATom and HIPPO. What is really nice is that the data from these airborne campaigns are compared with ozone sonde measurements (when possible) and data from IAGOS in the North Atlantic (civil aircraft measurements). I have no major comments to make except to congratulate the teams involved for the wonderful airborne missions conducted under the scientific leadership of Steve Wofsy. The paper is very clearly written with appropriate references and justifications of the statements that are made. The methodology is clearly stated. The results are well presented and the discussion is clear.**
**I would have wished to see a bit more background material in the introduction, and maybe this can be added. What did we know about ozone above the two oceans from previous experimental studies including space observations and ozonesondes? What did these studies show regarding ozone in the tropics and in the extra-tropics as a function of season? What did we know about the influence above the oceans of African and American biomass burning and lightning? What about the plumes from industrialized countries? And at the end of the paper, does the study confirm what was known or is there any new findings that would change our understanding of the processes involved? Perhaps a few sentences on these issues in the introduction and the conclusions would make the paper more attractive**

Response #1: We would like to thank the reviewer for the comment. We agree that the manuscript would benefit from more background context in the introduction and conclusions. We added a whole new paragraph in the introduction that provides contextual knowledge on the current understanding of $O_3$ distribution and climatology over the Pacific and Atlantic Oceans, L.56-82:

[revised manuscript text omitted]

L.594-596:

"In addition, ATom and HIPPO in situ measurements are invaluable to establish the quantitative legacy of pollution transport and chemistry globally by looking at the covariation of key species – in this case $O_3$ and CO."

**Reviewer 2**
**This paper reports tropospheric ozone distributions measured in ATom and HIPPO, compares them to ozonesondes and IAGOS to determine the consistency between the air sampled by ATom/HIPPO and long-term climatology, and comments on different features of the ozone distributions including vertical and seasonal distributions in the tropics and extratropics, and differences between the Pacific and Atlantic. Ozone curtain data from ATom and HIPPO are worth reporting in the literature, and it is also of interest to place ATom and HIPPO in a broader climatological context, as can be done with ozone data but has broader implications for the ensemble of ATom/HIPPO measurements for which such long-term records don't exist. This is a useful though limited contribution.**
**However, a big problem with the current paper is that it claims to discover what are very well-known features of the global ozone distribution, and it fails to credit previous work (notably from Jennifer Logan, Ed Browell's group, and many model papers) that made exactly the same points 1-3 decades ago. GTE campaign papers from Ed Browell's group using DIAL (PEM series, INTEX-B, SONEX, TRACEA, TRACE-P...) report very similar ozone curtains as ATom and HIPPO. All the findings reported here in the Abstract and Conclusions can be found in the previous literature. Once this is corrected and previous literature is properly credited, then it is not clear what is actually new in this paper in terms of scientific findings. It represents a limited contribution, and whether this is worth publishing in ACP is an Editor's decision.**

Response #2: We would like to thank the reviewer for the time taken evaluating our manuscript. We understand the issue raised by the reviewer regarding acknowledgment of prior work dealing with $O_3$ distribution over the Pacific and Atlantic Oceans. We have added a whole paragraph in the introduction that provides contextual knowledge on the current understanding of $O_3$ distribution and climatology over the Pacific and Atlantic Oceans, as suggested by Reviewer 1 (L.56-82, or see Response #1). References in this paragraph are in part related to the GTE campaigns, as suggested by the reviewer 2.

However, we want to point out that
i)     the aim of this paper is not to provide a review of all the work that has been done regarding tropospheric $O_3$ distribution. This would be a daunting, yet certainly useful, exercise that is not the goal here. We had rather opted for the option to provide adequate examples of literature that have previously shown similar, or different, results than found in our study (and acknowledged by the reviewer 1). In any case, our intent was certainly not to claim that most of our results are new findings. Therefore, we have changed the phrasing in the manuscript wherever it might have seemed ambiguous (e.g., L.19, 22, 30, 96-97, 395-397, 552, 567-568, 570-575, 580-581), and hope that this new version of the manuscript will meet the reviewer's transparency standards. As stated above, we also added more literature references in the introduction to acknowledge prior work and provide a more expanded contextual background to the study (see Response #1).
ii)    we believe that this paper will be an important contribution to the literature, for several reasons.
       First, evaluating the representativeness of in situ observations from airborne campaigns by comparing with longer-term observational records is a critical exercise, never achieved before to this extent (i.e., global-scale coverage). We think our paper will be a milestone in proving the usefulness of airborne observations to establish an $O_3$ - and other similarly long-lived species -

climatology, that have thus far been inferred from spatially-limited ozonesondes (Derwent et al., 2016; Diab et al., 2004; Liu et al., 2013; Logan, 1985; Logan and Kirchhoff, 1986; Oltmans et al., 2001; Parrish et al., 2013, 2016; Thompson et al., 2012) and commercial aircraft (Clark et al., 2015; Cohen et al., 2018; Kumar et al., 2013; Logan et al., 2012; Petetin et al., 2016; Sauvage et al., 2006; Zbinden et al., 2013), or from satellite and modeling work that has higher uncertainties (Edwards et al., 2003; Fishman et al., 1990, 1991; Hu et al., 2017; Thompson et al., 2017; Wespes et al., 2017; Ziemke et al., 2005, 2006, 2017). In addition, ATom and HIPPO in situ measurements help to establish the quantitative legacy of global pollution transport and chemistry by through the evaluation of key, covarying species – in this case $O_3$ and CO, as added L.594-596.

Second, Reviewer #2 suggests no new findings result from this analysis. An important distinction to make here is that most studies reporting global $O_3$ distribution result from satellite observations (e.g., Edwards et al., 2003; Fishman et al., 1990, 1991; Thompson et al., 2017; Wespes et al., 2017; Ziemke et al., 2005, 2006, 2017), modelling studies (e.g., Hu et al., 2017; Young et al., 2018) or from observations spatially expanded using back trajectory calculations (e.g., Liu et al., 2013; Tarasick et al., 2010). While useful, these studies come with somewhat large uncertainties, as recently noted by reports from the Tropospheric Ozone Assessment Report (TOAR), and thus require *in situ observations* to be used as a validation bench-mark (Tarasick et al., 2019; Young et al., 2018). ATom and HIPPO datasets provide exactly that, and therefore confirm and extend our understanding of global tropospheric $O_3$ distribution, sources, and processes, as noted L.586-590.

Third, there are several results highlighted in this paper that significantly extend tropospheric $O_3$ state of knowledge. For instance, our finding of a longitudinally similar $O_3$ distribution in the mid- and high-latitudes of both hemispheres has been disputed previously in the literature. The similarity of the distribution has been suggested for the northern mid-latitudes (Logan, 1985; Parrish et al., 2020) but it is not always evident in satellite-, modelling-, or ozonesonde-derived maps (e.g., Gaudel et al., 2018; Hu et al., 2017; Liu et al., 2013; Tilmes et al., 2012; Ziemke et al., 2017). Our observations show that this result mostly holds true for each season in the remote free troposphere (see section 4.2 of the manuscript). Another example is the $O_3$ to CO scatterplots shown in Figure 13, that highlight the wide-spread, year-round influence of continental outflow on $O_3$ in the remote troposphere in both oceans, and at almost all latitudes. Other studies have shown this for more spatially and temporally limited environments (e.g., Bey et al., 2001; Heald et al., 2003; Jaffe et al., 2004; Lin et al., 2017; Liu et al., 2002; Martín et al., 2006; Trickl et al., 2003), but our results show in situ evidence of frequent, large-scale occurrence of such events.

**Specific comments:**
**1. Abstract, lines 20-22: it is well known that there is more ozone in NH than SH.**

Response #3: We do not claim here that this is fundamentally new knowledge. However, ATom and HIPPO datasets are the first in situ observations covering scales large enough to actually confirm the modeling and satellite works that have shown this hemispheric gradient. We added a precision to the abstract, L.21-23:

"We highlight a clear hemispheric gradient, with greater ozone in the northern hemisphere consistent with greater precursor emissions, on par with previous modeling and satellite studies."

**2. Abstract, lines 22-23: uniformity of free tropospheric ozone at northern mid-latitudes has been known since Logan JGR 1985 from ozonesonde data.**

Response #4: We thank the reviewer for the reference. The reference has been added in the discussion where appropriate (L.487). However, our analyses expand beyond the northern midlatitudes, and provide the first in situ observation-based evidence of similar $O_3$ distribution longitudinally at the mid- and high latitudes of both hemispheres. Therefore, we would like to keep this sentence as is.

**3. Abstract, line 24: the higher ozone over tropical Atlantic than Pacific is well known – there is extensive literature on this from SHADOZ, TRACE-A, satellite retrievals (Jerry Ziemke)**

Response #5: Again, we do not claim to be the first to have found these results. That is why we chose to use the word "well-documented" (L.26). However, we have added substantial background context in the introduction regarding the high $O_3$ over tropical Atlantic citing appropriate references from GTE campaigns, ozonesonde networks, and satellite analyses (L.61-75):

"and the marked influence of African and South American biomass burning on $O_3$ production in the Southern Hemisphere (Browell et al., 1996b; Fenn et al., 1999; Mauzerall et al., 1998; Singh et al., 1996a; Thompson et al., 1996). Ozonesondes have been launched from remote sites for more than three decades in some places, and have provided additional constraint on the sources and photochemical balance of tropospheric $O_3$ including a deep understanding of vertically-resolved tropospheric $O_3$ climatology in select locations (Derwent et al., 2016; Diab et al., 2004; Jensen et al., 2012; Liu et al., 2013; Logan, 1985; Logan and Kirchhoff, 1986; Newton et al., 2017; Oltmans et al., 2001; Parrish et al., 2016; Sauvage et al., 2006; Thompson et al., 2012). Spatially-resolved $O_3$ climatology have been provided by commercial aircrafts routine sampling, but mostly limited to the upper troposphere or over continental regions (Clark et al., 2015; Cohen et al., 2018; Logan et al., 2012; Petetin et al., 2016; Sauvage et al., 2006; Thouret et al., 1998; Zbinden et al., 2013), and by satellite observations (Edwards et al., 2003; Fishman et al., 1990, 1991; Hu et al., 2017; Thompson et al., 2017; Wespes et al., 2017; Ziemke et al., 2005, 2006, 2017), somewhat tempered by large uncertainties (Tarasick et al., 2019)."

**4. Abstract, lines 26-28: Continental influences over the NH oceans have been known for decades (PEM-Tropics B, INTEX-B over Pacific come to mind), vertical structure is a well-known feature from ozonesonde and GTE data and again this has been extensively discussed in literature including countless model papers.**

Response #6: We do not claim to be the first to monitor continental influences on the ocean atmosphere. However, our study does provide the first in situ observation-based $O_3$ climatology over the Pacific and Atlantic Oceans in both hemispheres. We would also like to emphasize that East Asia's economy is completely different today than it was back in the 1990s. So, results from the 1980s and 90s may or may not hold true today.

Modelling work has been useful to advance the scientific community's understanding of $O_3$ seasonal cycles in places where measurements were lacking or limited. Models can now be evaluated using ATom and HIPPO dataset, not only looking at $O_3$ distribution, but also at $O_3$ seasonality, a significant advancement. This is now noted L.98-100:

"In addition, ATom and HIPPO sampling strategies were designed to deliver an objective climatology of key species to enable modelling of air parcel reactivity of the remote troposphere (Prather et al., 2017)."

**5. Abstract: that last sentence is gratuitous.**

Response #7: We modified the last sentence (L.29-32):

"This new dataset provides additional constraints for global climate and chemistry models to improve our understanding of both ozone production and loss processes in remote regions, as well as the influence of anthropogenic emissions on baseline ozone."

**6. Introduction: there has to be some serious review of previous knowledge on the tropospheric ozone distribution from ozonesondes, GTE aircraft campaigns, satellites, models. The authors are addressing a very old problem, on which there is a lot of literature. Ignoring that literature is not right, particularly because it leads to claims in the Abstract and Conclusions that suggest that the authors are unaware of it.**

Response #8: We have responded to this comment in Responses #1, #2, #3, and #4.

**7. Figures: there are way too many figures, and too much information on the figures – the eyes glaze, there's a lot of repetitiveness. It's hard to care about Sscore.**

Response #9: We think that 13 figures are well within the usual range of figures used in papers published in ACP, and that all are necessary to illustrate and support the discussion of our manuscript. If the reviewer could specifically recommend which figures are deemed gratuitous, we would consider changing or removing that figure. We would also like to point out that the $S_{score}$ is a necessary metric to provide a quantitative assessment of ATom and HIPPO representativeness, an important component of this paper, and this evaluation has been commended by Reviewer #1.

**8. Line 306: "airborne campaigns can capture global baseline O3 values". Didn't we already know this from the GTE campaigns?**

Response #10: The reviewer raises an interesting question. We argue that the comparison of airborne observations with longer-term observational records provides more statistically robust evidence that allows us to make this statement. In that sense, we think that no, we didn't already know that from the GTE campaigns, as an evaluation of GTE campaign representativeness has not been done.

**9. Lines 364-366, also 399-400, 409-411: we do know that ozone can be >100 ppb in biomass burning plumes over the South Pacific in spring (several papers coming out of PEM-Tropics A in Sep-Oct1996), seems like ATom/HIPPO just didn't hit them.**

Response #11: ATom and HIPPO did sample some BB plumes over the South Pacific in spring, as discussed L.512-514 and 539-545, and illustrated in Figure 13d.

**10. Lines 452-453 – uniformity of free tropospheric ozone at northern mid-latitudes has been known since Logan 1985 and has been shown in countless models.**

Response #12: Once again, our results are not limited to the northern mid-latitudes, but the mid- and high-latitudes of both hemispheres, a significant extension from the Logan et al. (1985) study. Nevertheless, we added this reference at L.487. If the reviewer could provide specific references for the models that have shown such results, we would certainly consider them. Regardless, ATom and HIPPO data provide the first in situ evidence of this pattern, and are therefore an important contribution to the $O_3$ literature.

**11. Lines 496-497: continental influence on ozone over the North Pacific is hardly new.**

Response #13: Agreed. That is why a full paragraph citing previous studies on the topic was included (L.500-507 in the original manuscript, now L.531-538). However, what is interesting and noteworthy from ATom and HIPPO datasets is the *year-round* influence of continental emissions on baseline ozone in the northern Pacific, indicated by elevated $O_3$ and CO in all seasons in Figure 13, whereas most of the existing literature focuses on spring and summer continental outflows (e.g., Heald et al., 2003; Jaffe et al., 2004; Lin et al., 2017).

**12. Lines 516-, Conclusions: the conclusions list the same unwarranted claims of "valuable new insight into O3 distributions in remote regions" (lines 520-521) as the Abstract. All of the features presented here about the global ozone distribution have been amply documented in the literature.**

Response #14: We have modified the text L.551-552:

"The data highlight several regional- and large-scale features of $O_3$ distributions, and provide  insight into $O_3$ current distributions in remote regions. The main findings are as follows:"

However, we disagree with the reviewer's statement "All of the features presented here about the global ozone distribution have been amply documented in the literature". Please refer to Response #2.

[revised manuscript text omitted]

Kumar, A., Wu, S., Weise, M. F., Honrath, R., Owen, R. C., Helmig, D., Kramer, L., Val Martin, M. and Li, Q.: Free-troposphere ozone and carbon monoxide over the North Atlantic for 2001–2011, Atmospheric Chemistry and Physics, 13(24), 12537–12547, doi:https://doi.org/10.5194/acp-13-12537-2013, 2013.

Lin, Horowitz Larry W., Cooper Owen R., Tarasick David, Conley Stephen, Iraci Laura T., Johnson Bryan, Leblanc Thierry, Petropavlovskikh Irina and Yates Emma L.: Revisiting the evidence of increasing springtime ozone mixing ratios in the free troposphere over western North America, Geophysical Research Letters, 42(20), 8719–8728, doi:10.1002/2015GL065311, 2015.

Lin, M., Horowitz, L. W., Payton, R., Fiore, A. M. and Tonnesen, G.: US surface ozone trends and extremes from 1980 to 2014: quantifying the roles of rising Asian emissions, domestic controls, wildfires, and climate, Atmospheric Chemistry and Physics, 17(4), 2943–2970, doi:10.5194/acp-17-2943-2017, 2017.

Liu, G., Liu, J., Tarasick, D. W., Fioletov, V. E., Jin, J. J., Moeini, O., Liu, X., Sioris, C. E. and Osman, M.: A global tropospheric ozone climatology from trajectory-mapped ozone soundings, Atmospheric Chemistry and Physics, 13(21), 10659–10675, doi:10.5194/acp-13-10659-2013, 2013.

Liu, H., Jacob, D. J., Chan, L. Y., Oltmans, S. J., Bey, I., Yantosca, R. M., Harris, J. M., Duncan, B. N. and Martin, R. V.: Sources of tropospheric ozone along the Asian Pacific Rim: An analysis of ozonesonde observations, Journal of Geophysical Research: Atmospheres, 107(D21), ACH 3-1-ACH 3-19, doi:10.1029/2001JD002005, 2002.

Logan, Staehelin J., Megretskaia I. A., Cammas J.-P., Thouret V., Claude H., De Backer H., Steinbacher M., Scheel H.-E., Stübi R., Fröhlich M. and Derwent R.: Changes in ozone over Europe: Analysis of ozone measurements from sondes, regular aircraft (MOZAIC) and alpine surface sites, Journal of Geophysical Research: Atmospheres, 117(D9), doi:10.1029/2011JD016952, 2012.

Logan, J. A.: Tropospheric ozone: Seasonal behavior, trends, and anthropogenic influence, Journal of Geophysical Research: Atmospheres, 90(D6), 10463–10482, doi:10.1029/JD090iD06p10463, 1985.

Logan, J. A. and Kirchhoff, V. W. J. H.: Seasonal variations of tropospheric ozone at Natal, Brazil, Journal of Geophysical Research: Atmospheres, 91(D7), 7875–7881, doi:10.1029/JD091iD07p07875, 1986.

Martin, B. D., Fuelberg, H. E., Blake, N. J., Crawford, J. H., Logan, J. A., Blake, D. R. and Sachse, G. W.: Long-range transport of Asian outflow to the equatorial Pacific, Journal of Geophysical Research: Atmospheres, 107(D2), PEM 5-1-PEM 5-18, doi:10.1029/2001JD001418, 2002.

Martín, M. V., Honrath, R. E., Owen, R. C., Pfister, G., Fialho, P. and Barata, F.: Significant enhancements of nitrogen oxides, black carbon, and ozone in the North Atlantic lower free troposphere resulting from North American boreal wildfires, Journal of Geophysical Research: Atmospheres, 111(D23), doi:10.1029/2006JD007530, 2006.

Mauzerall, D. L., Logan, J. A., Jacob, D. J., Anderson, B. E., Blake, D. R., Bradshaw, J. D., Heikes, B., Sachse, G. W., Singh, H. and Talbot, B.: Photochemistry in biomass burning plumes and implications for tropospheric ozone over the tropical South Atlantic, Journal of Geophysical Research: Atmospheres, 103(D7), 8401–8423, doi:10.1029/97JD02612, 1998.

Newton, R., Vaughan, G., Hintsa, E., Filus, M. T., Pan, L. L., Honomichl, S., Atlas, E., Andrews, S. J. and Carpenter, L. J.: Observations of ozone-poor air in the Tropical Tropopause Layer, Atmospheric Chemistry and Physics Discussions, 1–23, doi:10.5194/acp-2017-970, 2017.

Oltmans, S. J., Johnson, B. J., Harris, J. M., Vömel, H., Thompson, A. M., Koshy, K., Simon, P., Bendura, R. J., Logan, J. A., Hasebe, F., Shiotani, M., Kirchhoff, V. W. J. H., Maata, M., Sami, G., Samad, A., Tabuadravu, J., Enriquez, H., Agama, M., Cornejo, J. and Paredes, F.: Ozone in the Pacific tropical troposphere from ozonesonde observations, Journal of Geophysical Research: Atmospheres, 106(D23), 32503–32525, doi:10.1029/2000JD900834, 2001.

Pan, L. L., Honomichl, S. B., Randel, W. J., Apel, E. C., Atlas, E. L., Beaton, S. P., Bresch, J. F., Hornbrook, R., Kinnison, D. E., Lamarque, J.-F., Saiz-Lopez, A., Salawitch, R. J. and Weinheimer, A. J.: Bimodal distribution of free tropospheric ozone over the tropical western Pacific revealed by airborne observations, Geophysical Research Letters, 42(18), 7844–7851, doi:10.1002/2015GL065562, 2015.

Parrish, Law K. S., Staehelin J., Derwent R., Cooper O. R., Tanimoto H., Volz-Thomas A., Gilge S., Scheel H.-E., Steinbacher M. and Chan E.: Lower tropospheric ozone at northern midlatitudes: Changing seasonal cycle, Geophysical Research Letters, 40(8), 1631–1636, doi:10.1002/grl.50303, 2013.

[revised manuscript text omitted]

---

## Author Response (AR2)

Dear Editor,

Please find hereafter our response to your comment (changes to the manuscript are shown in red). The line numbers where those changes appear in the revised paper are also given at that point.

**Editor's comment:**

The reviewer #2 was questioning the novelty of this study. In the response, you have listed three points of the new contributions. I'd suggest you to consider including these discussions also in the manuscript, and revise your manuscript. Because of the nature of the reviewer's comments and revisions required, we may send the revised manuscript for further review.

Response to Comment:
Thank you for the suggestion. We considered adding all three arguments, detailed in the response to reviewers, in one paragraph, but we believed it did not really fit the flow of the paper. Instead, we chose to add the three points describing our new contribution throughout the manuscript, in places where we thought they best bolstered our arguments, as detailed below:

- The first point addressed the critical need for an evaluation of airborne in situ measurements by comparison with spatially collocated observations from well-established long-term monitoring networks. ATom and HIPPO missions, due to their extensive spatial and temporal coverage, allow for the first time (to the best of our knowledge) for such an exercise at various locations around the globe.
  We have highlighted this contribution first in the introduction:

  l.120-126: "Evaluating the representativeness of in situ observations from airborne campaigns by comparing them to longer-term observational records is a critical exercise never before done at such a global scale. We show that ATom and HIPPO measurements capture the spatial and, in some cases, temporal dependence of $O_3$ in the remote atmosphere, thus highlighting the usefulness of airborne observations to fill in the gaps of established but limited $O_3$ climatologies and other similarly long-lived species."

  And again, in the conclusion:
  l.574-575: "This representativeness evaluation on global scales highlights the usefulness of airborne observations to fill in the gaps of established but limited $O_3$ climatologies."

- The second point dealt with the novelty of having in situ measurements with global coverage to depict $O_3$ distribution rather than relying on useful, but imperfect satellite and modeling studies. Substantial discussion to this effect had already been added to the manuscript after responding to the reviewers' comments, but we emphasized even more on this aspect in the introduction, l.83-89:

  "Most studies reporting global $O_3$ distribution use satellite observations (Edwards et al., 2003; Fishman et al., 1990, 1991; Thompson et al., 2017; Wespes et al., 2017; Ziemke et al., 2005, 2006, 2017), modeling analyses (Hu et al., 2017), or observations spatially expanded using back trajectory calculations (e.g., Liu et al., 2013; Tarasick

et al., 2010). While useful, these studies come with somewhat large uncertainties, as recently noted by reports from the Tropospheric Ozone Assessment Report (TOAR), and thus require additional in situ observations to be used as a validation bench-mark (Tarasick et al., 2019b; Young et al., 2018)."

- The third point focused on several features described in our manuscript that we believe significantly confirm and extend our understanding of $O_3$ distribution and climatology, and the legacy influence of continental outflow on $O_3$ enhancements. One of these features is the similar tropospheric $O_3$ distribution observed year-round between the Atlantic and Pacific in the extra-tropics. This result has been highlighted in past and very recent studies in the mid-latitudes of the northern hemisphere, but disputed by other works. Discussion to this effect had already been added to the manuscript after responding to the reviewers' comments, but we emphasized even more on this aspect, l.494-496:

"However, the similarity of the $O_3$ distribution in the extra-tropical free troposphere above the Atlantic and Pacific is not always evident in satellite-, modelling-, or ozonesonde-derived maps (Gaudel et al., 2018; Hu et al., 2017; Ziemke et al., 2017)."

Another of these features is the wide-spread, year-round influence of continental outflow on $O_3$ in the remote troposphere in both oceans, and at almost all latitudes. This finding expands on a large body of literature have highlighted episodic and regional events of long-range transport of pollution plumes above the Atlantic and Pacific Oceans. Discussion to this effect had already been added to the manuscript after responding to the reviewers' comments, but we emphasized even more on this aspect, l.551-554:

"Our results expand on previous observation-based, but more spatially and temporally limited, studies that highlighted collocated enhancements of $O_3$ and CO at remote locations to show in situ evidence of frequent, large-scale influence of continental outflow on $O_3$ in the remote troposphere in both oceans, and at almost all latitudes."

And again, in the conclusion, l.604-608:

"In addition, ATom and HIPPO in situ measurements help to establish the quantitative legacy of global pollution transport and chemistry through the evaluation of key, covarying species – in this case $O_3$ and CO, and reveal the year-round pervasive influence of continental outflow on $O_3$ enhancements in the remote troposphere."

Sincerely,

Ilann Bourgeois, on behalf of the authors.